# Understanding the direct and indirect impacts of disease response phenotypes on chicken coccidiosis epidemiology: A modelling approach

Marie Ithurbide[1]*, Marie-Hélène Pinard van der Laan[1], Yuqi Gao[2,3], Andries D. Hulst[3], Mart C.M. De Jong[3], Andrea Doeschl-Wilson[4]

1 GABI, INRAE, AgroParisTech, Université Paris-Saclay, Jouy-en-Josas, France, 2 Animal Breeding and Genomics Group, Wageningen University & Research, Wageningen, The Netherlands, 3 Infectious Disease Epidemiology, Wageningen University & Research, Wageningen, The Netherlands, 4 The Roslin Institute, University of Edinburgh, Easter Bush, Roslin, Edinburgh, United Kingdom

* marie.ithurbide@inrae.fr

## Abstract

Coccidiosis, a widespread disease in poultry caused by protozoan parasites of the genus Eimeria, leads to significant economic losses. The increasing resistance of Eimeria species to anti-parasitics, combined with the high cost of vaccines, underscores the need for alternative intervention strategies against coccidiosis. This article explores the relative impact of several traits on the health of the group, accounting for the population dynamics of the infection. We focus on five traits that can potentially be influenced by genetic selection, treatment, vaccination or nutrition: (1) susceptibility, (2) recoverability, (3) infectivity, (4) tolerance, and (5) compensatory growth occurring after the infection ends. We propose an epidemiological model of coccidiosis based on literature review concerning chicken coccidiosis epidemiology and parameter estimations based on published data. Using this model, we investigate the direct and indirect impacts of each individual trait on the health and productivity of the flock. This approach aims at understanding the relative role of these individual traits on population level disease resistance and economical profitability of farms undergoing coccidiosis epidemics. The results showed increasing recoverability and tolerance were particularly beneficial for the health and productivity of the flock, both through direct and indirect effects whilst reducing infectivity has the highest beneficial effect on reducing the infectious load in the environment and on flock level protection. This approach has the potential to guide disease control strategies aimed at enhancing coccidiosis management within the poultry industry.

## Introduction

Coccidiosis represents one of the most economically significant diseases in commercial poultry production, causing substantial losses through reduced feed efficiency,

**Data availability statement:** All relevant data are within the manuscript and its Supporting Information files. The code generating the data is publicly available at: https://github.com/MarieIthurbi/coccidiosis_simulation.

**Funding:** This work was co-funded by the European Union's Horizon Europe Project 10113646 EUPAHW. The funders had no role in study design, data collection and analysis, decision to publish, or preparation of the manuscript.

**Competing interests:** The authors have declared that no competing interests exist.

decreased growth rates, increased mortality, and the costs associated with prevention and treatment [1,2]. Endemic in the majority of European poultry farms, this intestinal disease is caused by protozoan parasites of the genus *Eimeria*, which reproduce through a complex life cycle. Following ingestion of sporulated oocysts, sporozoites are released and invade intestinal epithelial cells where they undergo multiple rounds of asexual reproduction (schizogony) followed by sexual reproduction (gametogony), ultimately producing new oocysts that are shed in feces and contaminate the environment (Fig 1). The epidemiology of coccidiosis is influenced by multiple interacting factors including host genetics, environmental conditions, and management practices.

Several traits can be identified as influencing an individual's potential for disease transmission and its impact on flock health and farm profitability: 1- susceptibility, indicating the likelihood of infection upon contact with oocysts [3]; 2- recoverability, determining the duration of the infected and infectious period [4]; 3- infectivity, representing the average number of new infections caused by that infectious animal per unit of time [3]; 4- tolerance, defined as its capacity to maintain growth during infection [5,6]; and 5- compensatory growth occurring after the infection ends [7,8].

Infectiousness, i.e., the ability of an infected host to transmit the pathogen to susceptible individuals, can be decomposed into two traits: infectivity and recoverability. Infectivity is defined as the probability per unit time that an infected individual transmits the infection to a susceptible individual upon contact. Recoverability determines the rate in which an individual, once infected, recovers or dies from infection, thus relating to the duration an individual is infectious for [3]. Tolerance to infection, measured as the change in growth per unit change of within-host pathogen burden [9], is known to exhibit significant inter-individual and inter-breed variability. Compensatory growth following infection is an important aspect of disease resilience since it refers to the ability of a host to revert back to its performance levels in the absence of infection [7,8,10]. The capacity for compensatory growth after the development of immunity could partially offset the negative effects of early Eimeria infection, suggesting another potential target for disease management improvement [11–13].

Unlike conventional disease management programs where target traits are directly measurable, epidemiological traits such as susceptibility, infectivity, recoverability as well as tolerance are difficult to measure individually at scale. Instead, epidemiological traits manifest themselves through population-level infection patterns [4]. Moreover, while traditional genetic selection assumes that an individual's disease status depends on its own genetic make-up and the environment, epidemiological phenotypes also depend on the susceptibility, infectivity and recovery genes of other individuals in the contact group. Hence disease management often only focuses on direct benefits of traits, i.e., the benefit of one individual to have a certain trait, and miss the potential of indirect benefit (or costs) of traits on the health status of the entire flock/ herd [3].

Existing knowledge of coccidiosis epidemiology provides a foundation for modelling the effects of these traits on disease spread and impact. The objective of the present study is to develop an epidemiological coccidiosis model that incorporates

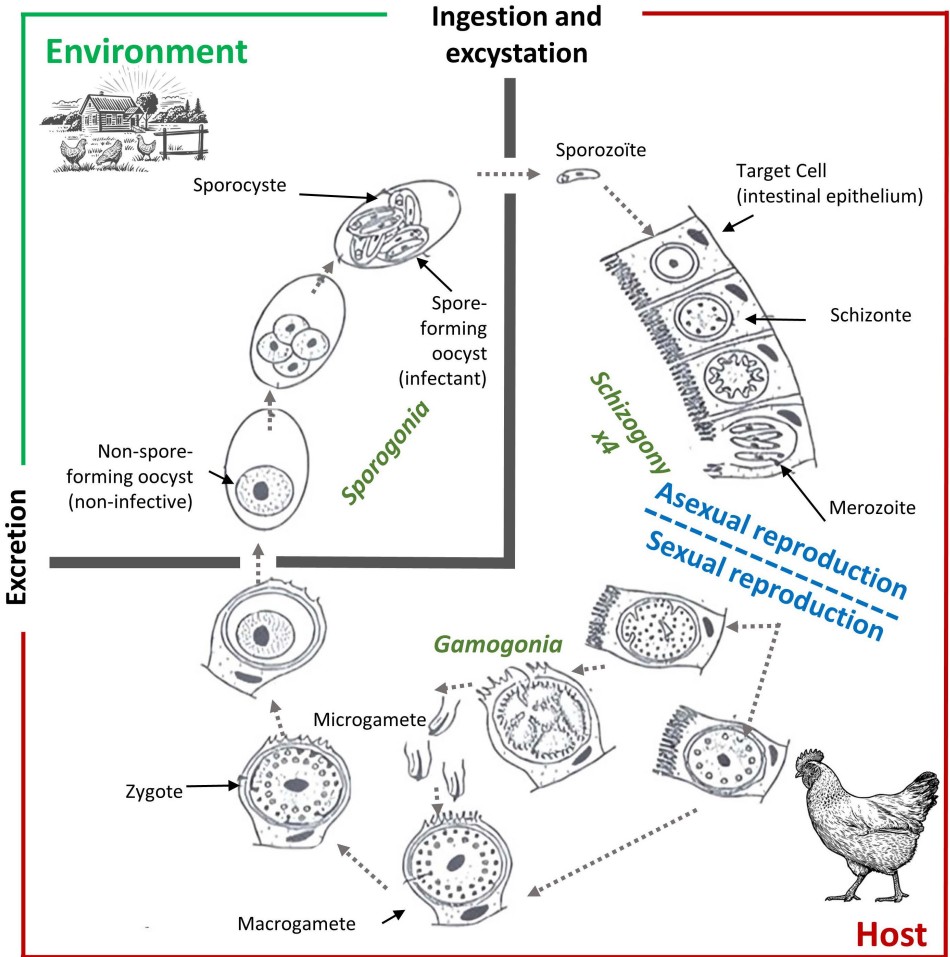

**Fig 1. Reproduction cycle of Eimeria sp.** A chicken becomes contaminated by ingesting a sporulated oocyst. This oocyst contains 8 sporozoites which will be released into the intestinal lumen after digestion of the oocyst shell. The sporozoites attach to epithelial cells and penetrate them. Then there is a schizogony phase, which is a phase of asexual multiplication inside the epithelial cell. There are 4 cycles of schizogony. After these 4 multiplication cycles, tens of thousands of merozoites are released in the intestinal lumen. During the asexual reproduction phase, the animal does not excrete pathogens in the environment. After this first phase that takes between 4 to 7 days, there is a phase of sexual reproduction, with production of female and male gametes which fuse to give an egg. The fertilized product forms a zygote which will be surrounded by a thick wall to protect it from the external environment. This oocyst is then excreted into the external environment. At the moment it is released into the external environment, it is not yet infective to a new host; it will become infective after sporulation or sporogony, 2 to 3 days after excretion into the environment. The oocyst is very resistant and can survive up to several months in the environment.

these 5 host traits and allows for estimation of key model parameters from existing experimental data. We then use the calibrated model to simulate epidemics by varying each host trait separately to determine the relative role of these individual traits on population level disease resilience defined as the ability of a herd/ flock to maintain high production performance when challenged by infection [3]. Moreover, we aim at estimating the relative impact of direct and indirect effects of these five traits on flock resilience. This knowledge could guide breeding programs and management strategies, leading to more targeted and effective methods for managing coccidiosis in the poultry industry.

## Materials and methods

In this section, we will first present the foundations of our epidemiological model. Then, we will explain how the parameters of this model were estimated from published data. Finally, we will demonstrate how we used this model to simulate different scenarios to understand the role of each of the 5 individual traits considered (host heterogeneity for susceptibility, infectivity, recoverability, tolerance, and compensatory growth) in flock resilience to coccidiosis and to estimate the relative weights of direct and indirect protective effects of these traits.

### The epidemiological coccidiosis model

A classical compartmental SEIR model [14,15] was used to simulate how coccidiosis spread through a population where individuals share a common environment. Assuming that some individual may die and assuming that recovered individuals become immediately susceptible again, we denote the model as SEID model, where everyone in the population falls into one of four categories: susceptible to infection ($S$), exposed ($E$), infected and infectious ($I$), or dead ($D$) (**Fig 2**). The total number of hosts, $N$, is constant across time $t$, and is represented by: $N(t) = S(t) + E(t) + I(t) + D(t)$. Since coccidiosis spreads through environmental transmission, we included an environmental compartment $L(t)$ where pathogens accumulate and decay (Chang & de Jong, 2023). Our model incorporates two additional factors: how infection affects the chicken's ability to gain weight, and in turn, how this reduced growth rate influences mortality among infected birds.

The epidemiological model equations are presented in Tables 1–2 and Table 3. In the case of environmental transmission, the force of infection experienced by one susceptible naïve individual is given by $\Lambda(t) = \beta L(t)/A$, which is the product of the average transmission parameter $\beta$ scaled by the individual's susceptibility, and exposure at time $t$, i.e., density of infectious load in the environment: $L(t)/A$ as expressed in [1] of Table 1. Our transmission formulation is

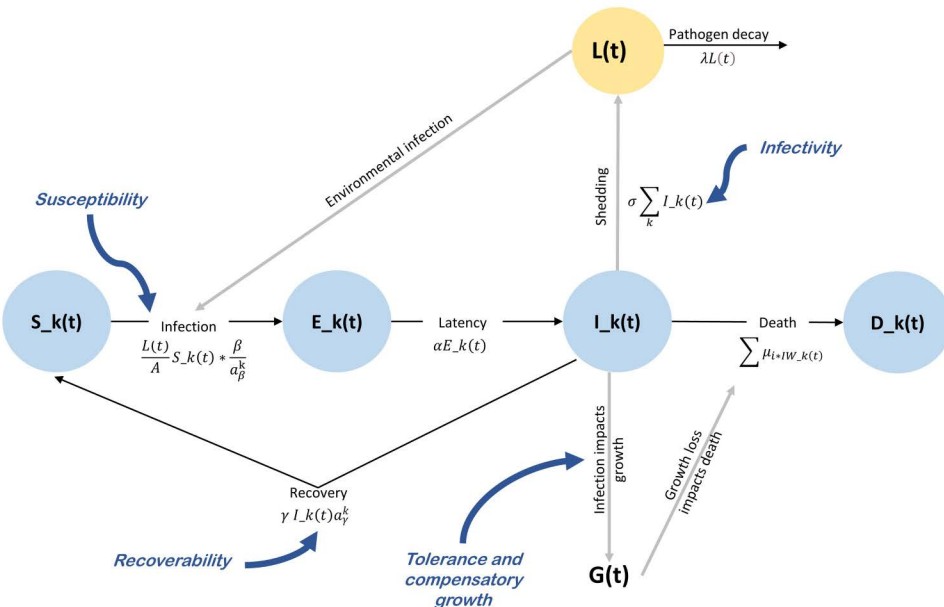

**Fig 2. Compartmental model of disease dynamics and growth impacts in animals.** The compartments are represented by circles: **S** for susceptible animals that can become infected, **E** for exposed animals that have been infected but are not yet infectious,   for infected animals capable of transmitting the disease, and **D** for animals that died from the infection. The model includes an environmental component showing pathogen concentration in the environment **L** and a growth G component representing animal growth trajectories affected by infection. The host traits impacting the different rates are represented in blue: susceptibility, infectivity, recoverability, tolerance and compensatory growth.

**Table 1. Differential model equations.**

| EPIDEMIOLOGICAL MODEL |
| --- |
| **Susceptible population ($S$)** |
| [1] $\frac{dS\_k(t)}{dt} = +\gamma\, I\_(k-1)(t)a_\gamma^k - \frac{L(t)}{A}S\_k(t) * \frac{\beta}{a_\beta^k}$ |
| **Exposed population ($E$)** |
| [2] $\frac{dE\_k(t)}{dt} = +\frac{L(t)}{A} * \frac{\beta}{a_\beta^k} * S\_k(t) - \alpha E\_k(t)$ |
| **Infectious population ($I$)** |
| [3] $\frac{dI\_k(t)}{dt} = +\alpha E\_k(t) - \sum_k \left(\gamma a_\gamma^k Ik(t) + \mu_i IW\_k(t)\right)$ |
| **Dead population ($D$)** |
| [4] $\frac{dD(t)}{dt} = +\mu_i \sum_k IW\_k(t)$ |
| **Infectious load in the environment ($L$)** |
| [5] $\frac{dL(t)}{dt} = +\sigma \sum_k I\_k(t) - \lambda L(t)$ |
| **Growth rates of individual chicken** |
| **Growth in the absence of infection** |
| [6] $\frac{dW^U}{dt} = G$ |
| **Growth under infection** |
| [7] $\frac{dW}{dt} = G(1-\zeta)$ |
| **Compensatory growth after infection** |
| [8] $\frac{dW}{dt} = G\left(1 + \rho\left(1 - \frac{W(t)}{W^U(t)}\right)\right)$ |

**Table 2. Definition of model's variables and initial values.**

| Variable | Definition | Initial value |
| --- | --- | --- |
| $S\_k(t)$ | Number of susceptible individuals at time t where k = 0,1,2 denotes the number of infections that individuals have already undergone. | 19 |
| $E\_k(t)$ | Number of exposed individuals at time t where k = 0,1,2 denotes the number of infections that individuals have already undergone. | 1 |
| $I\_k(t)$ | Number of infectious individuals at time t where k = 0,1,2 denotes the number of infections that individuals have already undergone. | 0 |
| $IW\_k(t)$ | Number of infectious individuals whose weight is below the survival threshold $G^*$ at time t where k = 0,1,2 denotes the number of infections that individuals have already undergone. | |
| $D(t)$ | Number of dead individuals at time t. | 0 |
| $W^U(t)$ | Unperturbed theoretical body weight at time $t$ (Sus) | 140 |
| $W(t)$ | Actual body weight at time $t$ (Sus) | 140 |
| $L(t)$ | Infectious load in the environment | 0 |

density-dependent: the infection probability scales with pathogen load per unit floor area (L(t)/A), such that higher stocking densities (more birds per m², i.e., smaller A for a given flock size) result in higher per-capita infection rates due to increased contact with contaminated litter. For animals undergoing multiple infections, acquired immunity reduces susceptibility. In our model it is implemented by the parameter $a_\beta^k$ that decreases the susceptibility of birds by a factor of $a_\beta$ each time it recovers from a new infection and $n_i$ increases. Hence, the transmission parameter is given by $\beta_k = \frac{\beta}{a_\beta^k}$ that has a different value for naïve animals ($k = 0$), or previously infected animals (k > 0).

**Table 3. Parameter names, values and descriptions.**

| Parameter | Value | Description | Source |
|---|---|---|---|
| $\beta$ | 1.61 m².d⁻¹ | transmission coefficient for naïve birds | A |
| $\alpha$ | ¼ d⁻¹ | Daily rate from exposed to infectious state | C |
| $\gamma$ | 1/12 d⁻¹ | Daily recovery rate | A |
| $\mu$ | 1 d⁻¹ | Daily mortality rate for individual with growth loss> $1 - G^*$. Value is 0 if growth loss <$1 - G^*$ | D |
| $\lambda$ | 1.15 d⁻¹ | Daily decay rate of infectious load in environment | A |
| $\sigma$ | 2.84 d⁻¹ | Mean daily shedding rate of infectious individuals | A |
| $a_\beta$ | 5.64 | Effect of acquired immunity on transmission rate | A |
| $a_\gamma$ | 2.3 | Effect of acquired immunity on recovery rate | A |
| $\zeta$ | 0.4 | Effect of infection on growth | B |
| $\rho$ | 0.1 | Speed of compensatory growth relative to logistic growth rate | D |
| $G^*$ | 0.8 | Threshold of relative weight below which animals die | D |
| $G$ | 22.5 g.d⁻¹ | Daily growth rate in the absence of infection | C |
| $A$ | m² | Area in which chickens are kept | |

A-Estimated from data published by Velkers et al.

B-Estimated from data published by Conway et al.

C-Values given by literature

D-Arbitrary choice

After ingestion of infectious oocysts, animals do not excrete oocysts for several days, corresponding to the exposed state ($E$). This prepatent period varies slightly among Eimeria species: approximately 4 days for E. acervulina [16–18], around 5 days (~121 hours) for E. maxima [17], and 4–5 days for E. tenella [19]. In our model, the average exposed period of individuals was assumed to be $1/\alpha$ = 4 days [2] of Table 1.

Throughout the infectious period, shedding rate of chicken will decrease due to development of immunity and eventually fade out. In our model, the mean duration of the first infectious period equals to $1/\gamma$ when $k = 0$. Due to the persistency of acquired immunity, the infectious duration is decreased by a factor $a_\gamma$ at each subsequent infection. Hence, recoverability equals to $\gamma a_\gamma^k$ in [3] of Table 1.

Severe coccidial infections can result in host mortality when parasitic burden exceeds physiological tolerance. In our model, mortality is triggered when body mass reduction reaches a critical threshold $G^*$ [4] in Table 1. Infected individuals with body mass reduction below the threshold die at a rate $\mu$. This mortality mechanism represents the sole interaction from growth dynamics toward the epidemiological processes in our model framework.

Chicken become infected from ingesting oocysts, which are shed in the environment by infectious individuals. We modelled the level of infectious load in the environment in equation [5] in Table 1. The quantity of infectious feces depends on the current number of infectious birds, their shedding rate $\sigma$, and the decay rate $\lambda$. We modelled the oocysts decay in the environment by an exponential decrease with a rate $\lambda$. We assume a constant oocyst decay rate, representing average survival under typical broiler house conditions, though oocyst survival can vary substantially with environmental conditions through time. [19] showed that the ingestion of unsporulated oocysts could lead to infection, and so we decided not to include the sporulation duration in our model.

We implemented the epidemiological model as an individual-based stochastic model using a modified version of the Doob-Gillespie algorithm [20]. Individual differences in infection outcomes arise from stochastic variations in each individual's infection state transitions, where the timing and probability of state changes (such as becoming infectious or recovering) vary stochastically according to the underlying epidemiological parameters.

## Growth of chicken

Growth loss is a main negative consequence of coccidiosis in chicken, even for subclinical coccidiosis when no other symptoms can be seen [13,21]. In the absence of infection, the theoretical growth curve of broiler chicken during the first months of life is roughly linear [6] in Table 1. During infection, growth is modelled according to equation [7] in Table 1 where growth rate is reduced by a rate $1 - \zeta$. The magnitude of growth loss $\zeta$ is partly influenced by the strain of Eimeria. Conway et al. (1993) showed that the same inoculation dose depressed weight gain by 10% for *E. acervulina*, by 23% for *E. maxima* and by 29% for *E. tenella*. After immunity has developed and the animal is no longer infected, a phase of compensatory growth may occur [11,13,21]. We modelled compensatory growth according to equation [8] in Table 1. The growth was expressed as the baseline growth rate ($G$) modified by a compensation term. This compensation term was proportional to the difference between the current weight of the animal and its expected weight in the absence of infection. The intensity of the compensatory response was controlled by a parameter $\rho$. This formulation allowed non-infected animals to grow faster than their baseline rate when their weight was below their expected trajectory (e.g., due to prior infection), with the growth rate returning to baseline as they approached their expected weight.

## Estimation of epidemiological parameter

**Transmission experiments.** Oocyst transmission data were obtained from previously published pairwise experiments by [22], which examined oocyst excretion and the transmission dynamics of *Eimeria acervulina* among broiler chickens. The study comprised four separate experiments (I–IV), involving a total of 42 pairs of chickens. In experiment I–IV, the chickens were inoculated with sporulated oocysts at doses of 5, 50, 500, or 50,000 oocysts, with 9, 17, 10, and 6 pairs assigned to each experiment, respectively. Within each pair, one bird was directly inoculated, while the other was exposed through contact. Fecal droppings were collected daily starting from 3 days post-inoculation (dpi) to 29, 30, 30, 26 dpi in experiment I–IV, respectively. A chick was classified as non-shedding for a given day only if both McMaster counting chamber technique (McM) and sedimentation–flotation (SF) technique returned negative results. Based on the days of shedding reported in this experiment, we assumed a latent period of 4 days: chicken were considered "exposed" for 4 days before the first shedding day. Chicken with three or more consecutive days of negative test results were classified as "susceptible", with susceptible period started from the first day of this negative period.

## Epidemiological model adjustment for the parameter estimation

Some adjustment of the epidemiological model presented in Table 1 was required for parameter estimation to match the experimental data and facilitate mathematical estimation. The parameters to be estimated are the transmission coefficient $\beta$, the effect of acquired immunity on transmission rate $a_\beta$, the decay rate $\lambda$, the shedding rate $\sigma$, the recovery rate $\gamma$, and the effect of acquired immunity on recovery rate $a_\gamma$. In [22], no deaths were recorded and no information on growth was available. Therefore, the adjusted model for parameter estimation included only the epidemiological components and did not incorporate growth modeling or a D compartment for dead animals.

In **Table 1**, the decrease of susceptibility and increase of recoverability of animals that overcame several infections was modelled via the parameters $a_\beta$ and $a_\gamma$ that corresponded to the effect of acquired immunity on transmission rate and recovery rate respectively. Like in **Table 1** the compartment S, E and I were divided in different compartments of homogenous animals. In [22] three infection rounds were detected at most. Therefore, in parameter estimation model, we specify 3 types of susceptible individuals ($S_0$, $S_1$, $S_2$) to capture the difference in immunity and susceptibility, 3 corresponding types of exposed individuals ($E_0$, $E_1$, $E_2$), and 3 corresponding types of infectious individuals ($I_0$, $I_1$, $I_2$). An additional S compartment, $S_3$, was included, however no events (new infection, transition, recovery/removal) involving chickens in the $S_3$ state were observed, so no further analysis will be applied on $S_3$. All data are available from the original publication [22]. After applying our assumptions, the revised datasets for each pair in experiments I–IV are presented in Suppl. 1, Datasets.

**Parameter estimation.** To estimate shedding rate $\sigma$, decay rate $\lambda$, and infection round i specific transmission rates $\beta_i$ in the above model from the experimental data, we applied the mathematical inference framework developed by Chang and de Jong [23]. The framework assumes that once infectious individuals are introduced into a clean environment, they shed pathogens at a specific rate, leading to pathogen accumulation in the environment, while the pathogens simultaneously decay exponentially at their natural decay rate. The environmental contamination level, which denoted as environmental infectious load $L(t)$ in this paper, is changing determined by the presence and shedding rate of infectious individuals at time $t$, as well as the pathogen's decay rate in the environment (Equation i).

$$\frac{dL(t)}{dt} = \sigma \cdot (I0(t) + I1(t) + I2(t)) - \lambda L(t) \tag{i}$$

With $L(0) = 0$, which represent that before experiments started, the environments were assumed to be totally clean.

The unit of $L(t)$ is defined differently from the inoculation dose of Oocysts (Oocyst counts). Instead, it is standardized in the transmission process: L(t) is modeled as a virtual epidemiological quantity representing effective transmission potential. This cannot be directly converted to oocyst counts because the relationship between oocyst numbers and infectivity is non-linear and depends on sporulation rates, environmental conditions, and immune status of the host population. If an infectious individual is placed in a clean environment for one time unit, the exposure $\int_0^1 L(t)dt$ equals to 1 unit. By solving $\int_0^1 L(t)dt = 1$ for the shedding rate $\sigma$, we can present $\sigma$ in terms of the decay rate $\lambda$ (Equation ii). The detailed calculation process can be found in (Chang & de Jong, 2023).

$$\sigma = \frac{\lambda^2}{-1 + e^{-\lambda} + \lambda} \tag{ii}$$

By entering equation ii for $\sigma$ into equation i, $L(t)$ can be further solved and simplified:

$$L(t + \tau) = \frac{(1 - e^{-\lambda\tau})\lambda}{-1 + e^{-\lambda} + \lambda} \cdot (I_0(t) + I_1(t) + I_2(t)) + e^{-\lambda\tau}L(t) \tag{iii}$$

The susceptible population is heterogeneous due to varying immunity levels. For chicken undergoing multiple infections, acquired immunity reduces their susceptibility. The differences among $S_0$, $S_1$, and $S_2$ are captured in the transmission rate parameter, $\frac{\beta}{a_\beta^k}$, which accounts for different susceptibility based on infection history $k$, with $k \in \{0, 1, 2\}$. Within each observed time interval $(t, t + \tau)$, the number of new infections for each type of susceptible can be modeled using a binomial distribution. The number of susceptible serves as the number of trials, and the corresponding infection probability defines the likelihood of a new infection. Take $S_0$ as an example, let $S_{t,t+\tau,k=0}$ represent the number of $S_0$ during $(t, t+\tau)$. The number of new infections in $S_0$, denoted as $C_{t,t+\tau, k=0}$, occur with infection probabilities $p_{t,t+\tau,k=0}$. To calculate the infection probability, the hazard rate at time $t$ is first determined by $\frac{\beta}{a_\beta^k}L(t)$. Next, exposure is calculated by integrating hazard rate over $(t, t + \tau)$. The infection probabilities $(p_{t,t+\tau,k})$ and the corresponding likelihood function $(\mathcal{L}(\theta))$ in all three susceptible $(S_0, S_1, S_2)$ are then given by the following equations:

$$p_{t,t+\tau,k} = 1 - e^{-\frac{\beta}{a_\beta^k} \cdot \frac{\int_t^{t+\tau} L(x)dx}{N_{t,t+\tau}}} \tag{iv}$$

$$\mathcal{L}(\theta) = \prod_{k=0}^{2} \binom{S_{t,t+\tau,k}}{C_{t,t+\tau,k}} (1 - p_{t,t+\tau,k})^{C_{t,t+\tau,nk}} p_{t,t+\tau,k} (S_{t,t+\tau,k} - C_{t,t+\tau,k}) \tag{v}$$

Where $\theta = \{\beta, \lambda, a_t\}$. $N_{t,t+\tau}$ represent the number of individuals in the pair, which equals 2. The AIC value can be obtained based on the Akaike equation for each parameter set $\{\beta, \lambda, a_t\}$. We obtained our Maximum Likelihood Estimation at the minimum AIC value. The 95% confidence interval (95%CI) were determined by applying the minimum AIC value plus 2 [24,25].

$$AIC = 4 - 2\widehat{\mathcal{L}(\theta)} \tag{vi}$$

To calculate the average recovery rate $\gamma$, which is defined as the inverse of the mean infectious period, we conducted analysis on the durations of infectious periods. The infectious periods were recorded across all experiments but were not distinguished between $I_0$, $I_1$, $I_2$. To account for potential heterogeneity among populations with varying infection histories ($k \in \{0, 1, 2\}$), the infectious periods for $I_0$, $I_1$, $I_2$ were analyzed separately (based on the dataset presented in Suppl. 1). Based on the infectious periods for $I_0$, $I_1$, $I_2$, a statistical model was developed to examine the dependence of $\gamma$ on $k$.

## Determination of growth parameters

The value for the parameter defining the growth rate in the absence of infection ($G$) was obtained from literature concerning slow growing broilers. Birth weight of chicken is around 30 g [26,27]. Weight at 80 days of age culling of slow growing Premium Hubbard broilers is around 1800g (https://www.hubbardbreeders.com/fr/produits/males-hubbard/7756-males-hubbard-premium-a-croissance-lente.html). Given a roughly linear growth, growth rate $G = 22.6$ g/day.

The value for impact of infection on growth (i.e., parameter $\zeta$ in equation [7]) was obtained from published challenge experiments. Conway et al. [28] studied the impact of several ingestion levels of *E. tenella*, *E. acervulina*, and *E. maxima* on 10-day-old male Hubbard chickens regarding their growth 5 days post-ingestion. This article highlights growth loss of up to 26.7% of live weight compared to the control group in cases of *E. acervulina* ingestion. They also demonstrate that at equal ingestion levels, weight losses are more significant for infections with *E. maxima* and *E. tenella*. This weight loss is rapid, given that the incubation period before oocyst excretion is 4 days. Other studies examine weight loss at 8 days post-infection with *E. tenella*. Summary estimates from the literature are depicted in (Table 4). The absence of repeated weight measurements in these studies makes it impossible to determine with certainty the weight loss patterns throughout a complete infection duration (10 days after the start of excretion). Moreover, weight loss appears to be multifactorial and depends on the chicken breed and sex considered, the oocyst ingestion dose, and the Eimeria species. We decided to take the value of 20% growth loss at 10 days post-infection for an E. acervulina infection. According to equation [7] in Table 1, infected chickens deviate on average by $(\zeta * G)$ g/day from their theoretical weight in the absence of infection. For 15-day-old chicks weighing approximately 450 g, a loss of 20% of live weight in 10 days corresponds to $\zeta = 0.4$.

Due to lack of data on compensatory growth after coccidiosis infections in chickens, we arbitrarily set the compensatory growth parameter $\rho$ to 0.1, representing a moderate capacity to recover lost weight through accelerated growth post-infection.

## Evaluation of the effect of individual traits on the health and performance of the flock

First, we investigated how decreased host susceptibility (decreased $\frac{\beta}{a_\beta^k}$), decreased infectivity (decreased $\sigma$), increased recovery rate (increased $\gamma a_\gamma^k$), increased compensatory growth (increased $\rho$) and increased tolerance (increased $\zeta$) affect flock performance and health. Specifically, for each trait separately we compared the health and performance of the flock in a baseline scenario (traits value in Table 3) and in the alternative scenarios where the value of the trait was set to a more favourable value with an increase or decrease of 1 ln unit. The scenarios were called Sus, Inf, Rec, CompG and Tol for susceptibility, infectivity, recovery rate, compensatory growth and tolerance respectively.

We ran 500 simulations in each scenario, with 20 individuals in a flock, and one bird getting infected at 5 days of age. Culling occurred at 84 days of age (slow growing chicken). Key metrics used to evaluate flock health and performance included the mean number of infections animals underwent, the mean prevalence, the mean infection duration per animal, the percentage of death, the mean and standard deviation of weight at culling (grams), the mean maximum weight deviation (in percentage), the mean sum of daily weight deviation per animal (grams).

Several articles [29–31] have proposed an innovative measure of resilience to disease based on the area of an individual's trajectory in the 2 dimensional "disease space," which is defined by internal pathogen load and health status.

**Table 4. Maximum weight deviation compared with non-infected animals across several articles after experimental ingestion of Eimeria oocysts.**

| Article | Chicken breed | Age at infection | Day at weighing | Eimeria strain | Infection load (n oocysts) | Max weight deviation (%) |
|---|---|---|---|---|---|---|
| Boulton.2018 | 4xCobb500 | 21 | 7 | *E.tenella* | 40000 | −15.3 |
| Conway.1993 | Hubbard | 10 | 5 | *E.acervulina* | 100 | 1.0 |
| | | | | | 1 000 | −2.9 |
| | | | | | 10 000 | −3.9 |
| | | | | | 100 000 | −10.4 |
| | | | | | 1 000 000 | −26.7 |
| | | | | *E.maxima* | 67 | 2.6 |
| | | | | | 670 | 2.9 |
| | | | | | 6 700 | −7.1 |
| | | | | | 67 000 | −23.9 |
| | | | | *E.tenella* | 100 | −0.1 |
| | | | | | 1 000 | −1.9 |
| | | | | | 10 000 | −11.0 |
| | | | | | 100 000 | −28.9 |
| Hamet.1982 | Fayoumi | 15 | 8 | *E.tenella* | 200 000 | −1.0 |
| | Fayoumi.Rhode.island | | | | | −10.5 |
| | Rhode.island M99 | | | | | −12.6 |
| | Rhode.island.Fayoumi | | | | | −10.5 |
| Pinard.1998 | Fayoumi | 28 | 8 | *E.tenella* | 150 000 | −10.8 |
| | Mandarah | | | | | −28.2 |
| | Rhode.island | | | | | −20.8 |
| | White.Leghorn WLB21 | | | | | −20.8 |
| | White.Leghorn WLDW | | | | | −28.6 |

During early infection, pathogen load increases while health decreases. As pathogen load subsequently decreases, the host begins actively repairing its health, creating an open loop where each position along the path represents a unique state. The most resilient animals are characterized by smaller loops, as they successfully limit both pathogen load and detrimental health effects. We propose extending this methodology to the population level, where population resilience to disease is defined by the size of its loop in the area bounded by environmental infectious load and the flock's mean weight deviation. Hence, a batch of chicken will be considered resilient to coccidiosis if the area of the loop is small, i.e., if the batch limits environmental infectious load L(t) and average weight deviation of animal with fast recovery. The area within the loop was calculated using Green's formula as demonstrated by [32], which provides a method to compute the area enclosed by a closed curve using line integrals. Statistical comparison of the different performance traits in the baseline scenarios with the different alternative scenario was performed using the Student t-test. The multiplicity of the test was addressed applying a Bonferroni correction to the p value.

### Estimating the direct and indirect effect of traits on individual health and performance

The positive (or negative) effects of the considered traits on health and production of the flock operate through two different underlying mechanisms. First, there is the direct effect: an individual with a positive trait will have better health and production because the trait protects itself. Second, there is the indirect effect: an individual's health or performance is affected by flock members' trait value (e.g., lower infectivity of flock members decreases exposure dose and thus health of

focal individual). In order to determine the direct and indirect protective effects of each trait on individual and flock health and performance, respectively, we compared the performance of one individual *i* in 3 situations:

- Baseline: Individual *i* has baseline trait value in a population with all individual having baseline trait value.

- Direct effect determination: Individual *i* has a favorable trait value (same scenarios as previous paragraph) in a population with all other individuals having a baseline trait value.

- Indirect effect of flock mates on individual: Individual *i* has baseline trait value in a population with all individuals having a favorable trait value (same scenarios as previous paragraph).

For each of the five traits, we performed 500 replicate simulations per scenario as well as for baseline scenario. Each simulation consisted of a group of 20 birds, with one individual becoming infected at 5 days of age. Direct and indirect effects were assessed by comparing individual with baseline performance in terms of: number of infections, duration of infectious time, time before first infection, death, maximum weight deviation, cumulated daily weight deviation and weight at culling. Statistical comparisons were performed using Student's t-tests. The multiplicity of the test was addressed applying a Bonferroni correction to the p value.

## Results

### Estimates of model parameter values

**Transmission, decay and shedding rate parameters.** Using the maximum likelihood function, the best estimates for three unknown parameters—transmission rate parameters ($\beta$ and $a_\beta$ in $\frac{\beta}{a_\beta^k}$) and the decay rate parameter ($\lambda$)—were obtained. Subsequently, the fourth unknown parameter, the shedding rate parameter ($\sigma$), was fixed based on the decay rate parameter (Equation ii). The detailed results and corresponding profile likelihood plots are presented in Table 5 and S1 Fig.

**Recovery rate.** A statistical model was developed to quantify the relationship between the recovery rate ($\gamma$, which calculated as 1/Infectious Period) and the number of infection histories ($n_i$). A logistic growth model was used, expressed as: $\gamma\_k = \frac{2.30^k}{12.1}$ d$^{-1}$ (95%CI: $\frac{1.61^k}{13.3} - \frac{2.99^k}{10.9}$) d$^{-1}$.

### Simulated infection and weight profiles

Fig 3 illustrates the outcome of 100 simulations involving 20 chickens each in a shared environment, demonstrating the dynamics of Eimeria infection over time. The simulation was initiated with one chicken infected at 5 days of age. No variability for individual traits was included. However, the 100 iterations of the simulation highlight the large variation due to stochasticity alone. The results reveal a clear pattern of infection spread, with the initially infected chicken causing a rapid increase of prevalence (i.e., proportion of individuals in E and I state), followed by a gradual decline. The trend in the environmental contamination (graph B) mimics the trend of individuals in the 'I' compartment. The weight curves are

**Table 5. Results of transmission rate, decay rate, shedding rate Parameters Estimation.**

| Parameter | Notation | Estimate | 95% CI |
|---|---|---|---|
| Transmission coefficient | $\beta$ | 1.61 d-1 | (1.27, 2.01) d$^{-1}$ |
| Effect of acquired immunity on transmission rate | $a_\beta$ | 5.65 | (4.33, 7.49) |
| Decay rate | $\lambda$ | 1.15 d-1 | (0.883, 1.51) d$^{-1}$ |
| Shedding rate | $\sigma$ | 2.84 d-1 | (2.63, 3.12) d$^{-1}$ |
| Duration of the infection period | $1/\gamma$ | 12.1 d | (10.9, 13.3) d |
| Effect of acquired immunity on recovery rate | $a_\gamma$ | 2.30 | (1.61, 2.99) |

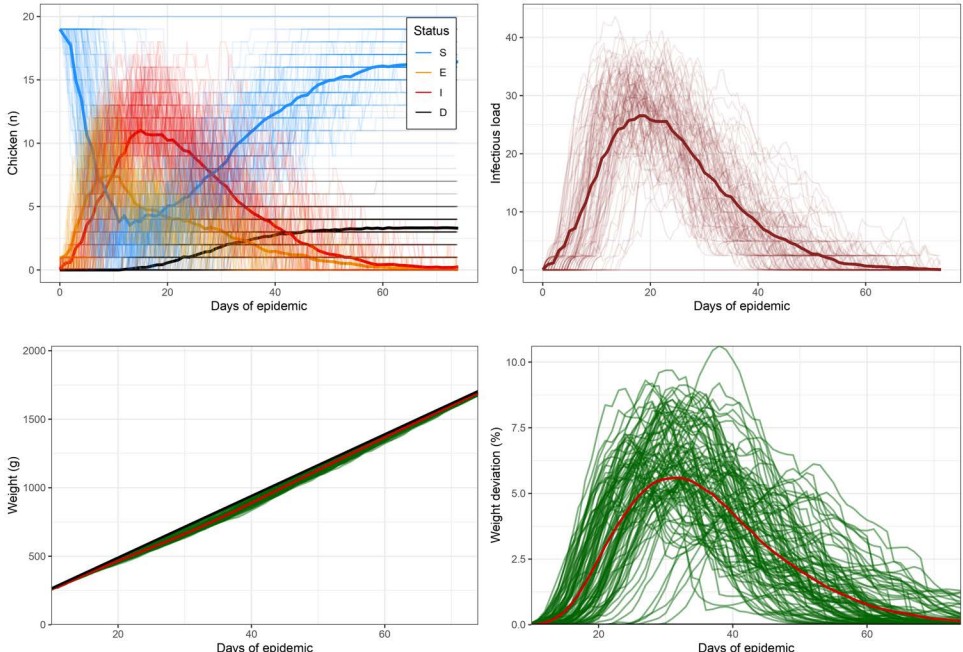

**Fig 3. 100 epidemics are simulated among 20 chickens in a shared environment.** At time = 0 (5 days of age), one chicken got infected. The figures display the infection status of each animal over time, the infectious load in environment, individual weight curves and weight loss (percentage compared to theoretical weight) from day 0 to day 74 after the beginning of the epidemic.

closely synched with I(t) and L(t), demonstrating the impact of infection on growth, with infected chickens showing temporary growth depression followed by compensatory growth.

## Model fit evaluation

Model fit was evaluated by comparing the time profiles of incidence rates for the predicted new cases with those of observed new cases from the transmission experiment by (reference), and by validating the estimated decay rate parameter against published oocyst survival data. In experiments I-IV, the infection probabilities for $S_0$, $S_1$, and $S_2$ in each pair from time 0 to time $t$ were calculated using Equation iv, with results depicted in **Fig 4**. This visualization highlights how well the model aligns with observed data across different experiments.

The relationship between the half-life ($t_{1/2}$) and the decay rate parameter ($\lambda$) is given by $t_{1/2} = \frac{\ln(2)}{\lambda}$, while the mean survival time ($\bar{t}$) for an oocyst can be calculated as $\bar{t} = \frac{1}{\lambda}$. Using our estimated decay rate, the half-life of oocysts was calculated as 0.601 d (95%CI: 0.460–0.785 d), and the mean survival time as 0.867 d (95%CI: 0.663–1.13 d). One study showed that oocysts begin to deteriorate after only 24 h and after five days, most oocysts were non-viable, but some could be detected up to day 23 [33]. It suggests that half-life of oocysts is larger than our estimate. However, our estimate refers to infectious load half-life that is not necessarily the same as oocysts half-life. Moreover, oocysts survival in environment and sporulation rate closely relate on environments parameters such as temperature and humidity [34].

As oocysts remain viable in the environment, diagnosed oocyst counts from infectious chickens may underestimate the environmental infectious load. To address this, our model quantifies the cumulative environmental infectious load as $L(t)$. Different from oocyst counts, which represent discrete observations at sampling points, $L(t)$ captures the continuous dynamics of oocyst accumulation and natural decay over time, resulting in a delayed peak and extended right tail (Fig 5). To note, the units differ, as $L(t)$ uses a scaled unit based on model assumptions instead of oocyst counts per gram of feces.

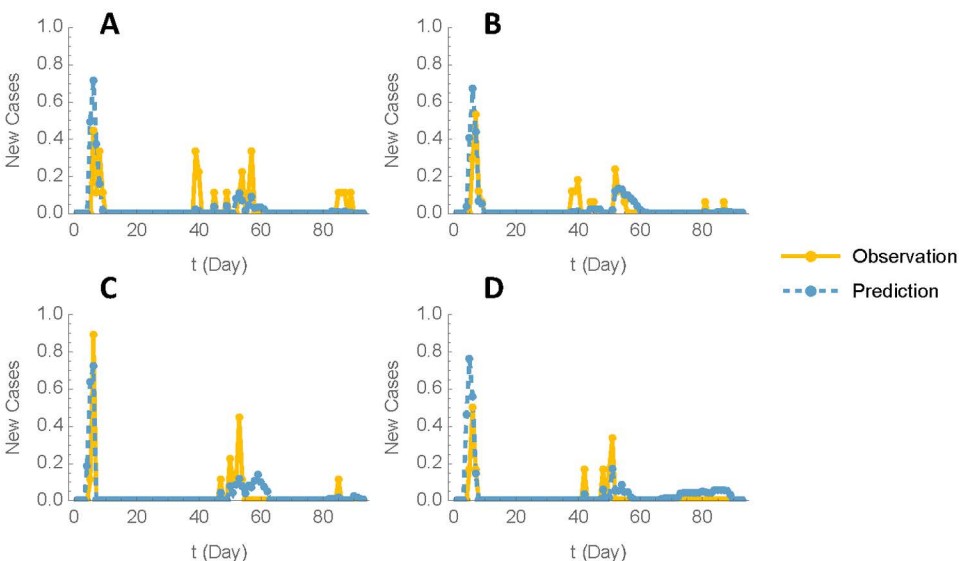

**Fig 4. Model Fit Evaluation.** Predicted incidence rates for new Cases vs. Observed incidence rates across Experiments I-IV in [22]. The plot compares predicted new cases (dashed lines) and observed new cases (solid lines) across experiments I-IV (Panel A-D).

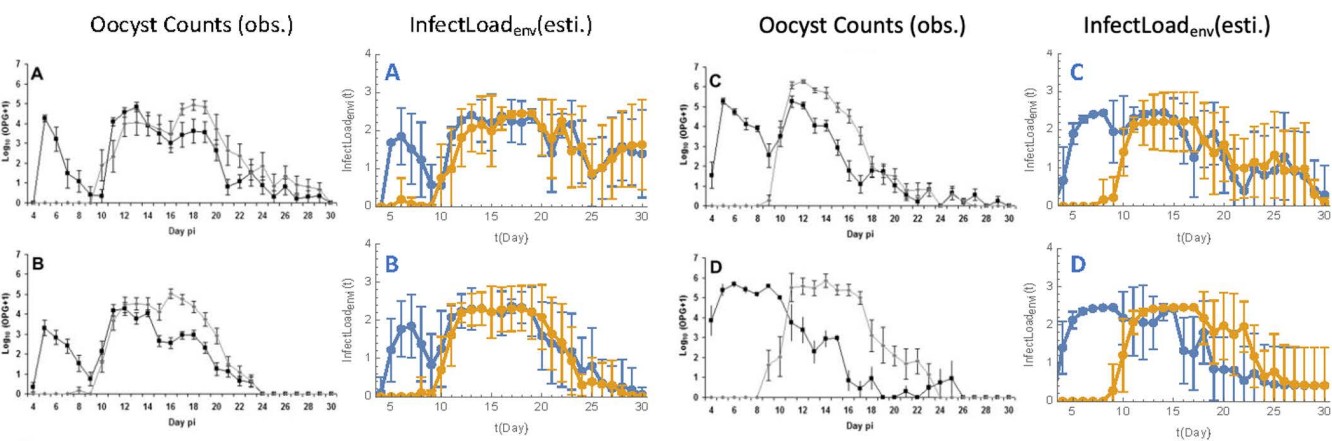

**Fig 5. Comparison of Observed Oocyst Counts and Predicted Environmental Infectious Loads L(t) across Experiments.** This Fig compares the dynamics of the environmental infectious load derived from our model (L(t), a dimensionless measure of transmission potential) with previously published results [35]. Panels A to D show the experiments I-IV. The black (left) and blue (right) lines both represent inoculated chicken, while grey (left) and yellow (right) lines both represent contact chicken. Error bars indicate variability between different pairs in each experiment. The x-axis shows time after inoculation (days), and the y-axis represents oocyst counts (left) and L(t) (right).

## Effect of the diverse host trait values on individual and flock health and growth performance

Fig 6 shows the outputs of the 3 000 simulations (500 simulations per scenario) in the baseline scenario and the 5 alternative scenarios comprising one improved host trait at a time: reduced mean infectivity by one ln unit (Inf), reduced mean susceptibility by one ln unit (Sus), increased mean recoverability by one ln unit (Rec), increased mean

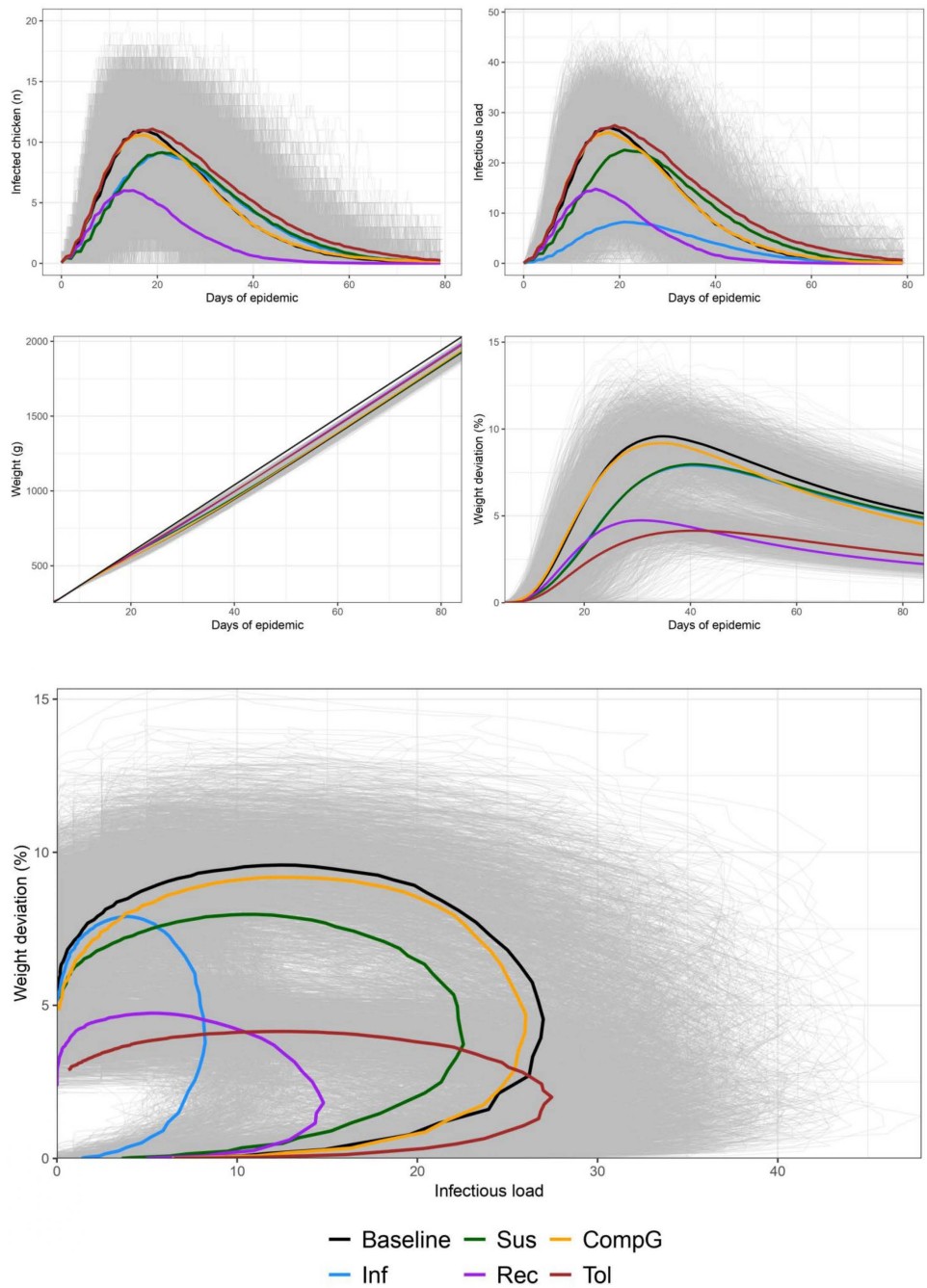

**Fig 6. Impact of host traits on disease dynamics and flock performance.** Comparison between baseline scenario and scenarios with modified traits: reduced infectivity (Inf), reduced susceptibility (Sus), increased recoverability (Rec), increased compensatory growth (CompG) and increased tolerance (Tol). For each scenario the corresponding parameter was increased or decreased of 1 ln unit compared with baseline. Grey lines represent individual simulation trajectories, colored lines show mean trajectories per scenario. Panels show from top left to bottom: infected chickens over time, infectious load in environment, weight growth curves, weight deviation percentage over age, and weight deviation percentage against infectious load.

compensatory growth by one ln unit (CompG) and increased mean tolerance by one ln unit (Tol). The dynamic of the prevalence shows a peak of 50% infecteds reached around 18 days post inoculation of the first chicken in the baseline scenario. Increased tolerance (Tol) and increased compensatory growth (CompG) scenarios show similar results. The increased recoverability scenario (Rec) did not change the timing of the peak, but decreased the magnitude to a prevalence of 25%. The reduced susceptibility (Sus) and infectivity (Inf) scenarios on the contrary, induced a more modest reduction of the maximum prevalence (around 40%) but also delayed the peak to 22 days after inoculation of the first chicken. Fig 7 and S5 Table show that the number of infections per animal were most reduced in scenario Inf and Sus (−18.05% each), followed by scenario Rec (−9.27%), while scenario CompG did not affect the number of infections. Interestingly, scenario Tol showed increased number of infections per animal (+14.63%). It results from increased survival from infection leading to more re-infections. The same was observed for the total duration of infectious time and the mean prevalence, except that the scenario Rec showed strongest impact (−59.91% and −52.00% respectively). The death rate was reduced in all scenarios, with strongest impact in scenarios Tol and Rec (−100% and −95.24% respectively).

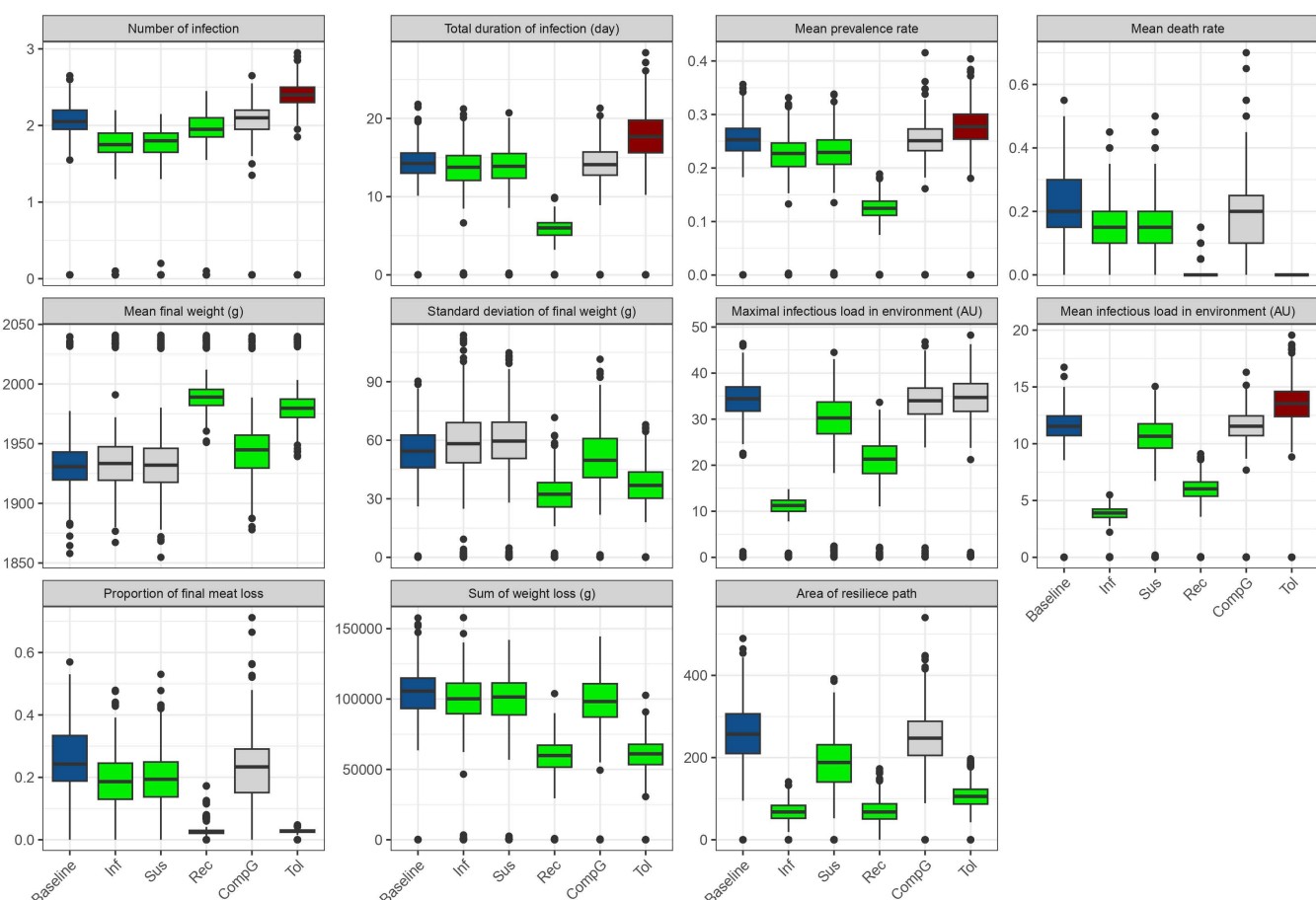

**Fig 7. Comparison of population average performances between baseline scenario and alternative scenarios.** Inf: decreased infectivity, Sus: decreased susceptibility, Rec: increased recoverability, CompG: increased compensatory growth, Tol: increased tolerance. The color of the boxplot corresponds to the associated p-values from t-tests comparing each scenario to baseline (grey: $p > 0.05$, green: $p < 0.05$ and favorable effect, red: $p < 0.05$ and deleterious effect). Simulations were performed with 500 replicates of 20 birds for each scenario.

The infectious load in the environment follows similar patterns to the prevalence dynamics. Scenario Inf shows the most dramatic reduction in environmental contamination, with a maximal infectious environmental load less than a third of the baseline scenario (−68.56%). This is followed by the scenario Rec (−39.98%) and the scenario Sus (−15.07%), while scenarios Tol and CompG maintain similar patterns to the baseline. The mean infectious load in the environment showed similar pattern, but with a detrimental effect of the scenario Tol (+15.43%), due as previously to the increased duration of infectious duration.

The weight and weight deviation percentage over age reveals distinct patterns of growth impact. In **Fig 6** the baseline scenario shows a maximum weight deviation of 9%. The increased tolerance scenario (Tol) demonstrates remarkable good impact with maximum weight deviation of 4%, being the lowest of all scenarios, despite having higher infection numbers resulting in a longer peak of weight deviation. **Fig 6** also shows that the increased recoverability scenario (Rec) also shows reduced weight impact, with deviation peaking at about 5%. Scenarios Inf and Sus show intermediate improvements, while the compensatory growth scenario (CompG) closely follows the baseline pattern with slightly faster compensatory growth following the peak of weight deviation. **Fig 7** shows that the final weight of animal reaching culling age and the homogeneity of the weight of the batch are favourably impacted in scenario Rec, Tol and CompG (+3.07% and +2.53% for weight at culling and −42.07% and −32.07% respectively for sd of the weight of the batch), with no impact in scenario Sus and Inf. The proportion of final meat loss followed strongly the death rate since it takes into account the expected sum of body weight at culling with 0% death. Hence the strongest impact of scenario Rec and Tol (0.03±0.03 and 0.03±0.01 respectively against 0.25±0.11 in baseline). The sum of weight loss for an animal reflects both the amplitude of the weight deviation and its duration. On an economical point of view, it roughly reflects the added feed consumption of the animal due to the disease. It is significantly impacted in all alternative scenarios, with strongest positive impact of the Rec and Tol scenarios (−88% each).

The resilience pathway (weight deviation vs. environmental infectious load) in the reduced infectivity scenario (Inf) produces the smallest loop on the x axis due to lower infectious loads, while the increased tolerance scenario (Tol) shows a compressed loop on the y axis despite higher infectious loads (**Fig 6**). The increased recoverability scenario (Rec) showed an intermediate loop due to both reduced infectious load and weight deviation due to reduced infectious time implying reduced shedding and reduced duration of weight loss. Scenario Sus showed more modest but balanced impacts on both weight deviation and infectious load. The CompG scenario did not show significant impacts on the resilience pathways. The comparison of the area of these loops shows strongest impact of the reduced infectivity (−74.02%) followed by increased recoverability (−73.26%), increased tolerance (−59.00%) and decreased susceptibility (−28.88%), with increased compensatory growth showing no significant decrease of the area of the pathway.

## Direct and indirect effect of traits on health and performance

Analysis of direct and indirect effects across traits reveals diverse patterns of individual performance as shown in **Fig 8** and supplementary Table 6. The decreased infectivity scenario (Inf) showed no significant direct effect in any measured parameters. However, it had several significant indirect favorable effects, with reduced average number of infections (−18.63%), maximum weight deviation (−16.29%), sum of weight loss (−14.92), and a delayed first infection (+44.18%) with reduced total infectious time (−8.33%) in animals with baseline traits when surrounded by individuals with reduced average infectivity by one exponential unit.

Decreased susceptibility (Sus) demonstrated significant direct effects, with reduced number of infections (−15.68%), reduced duration of infections (−9.72%), delayed time to first infection (+17.28%), reduced maximal weight deviation (−11.31%,) and reduced death rate (−27.28%) in individuals carrying the favorable trait value. However, no significant impact of the indirect effect was found suggesting that the presence of less susceptible individuals in the group did not substantially affect the performance of other animals.

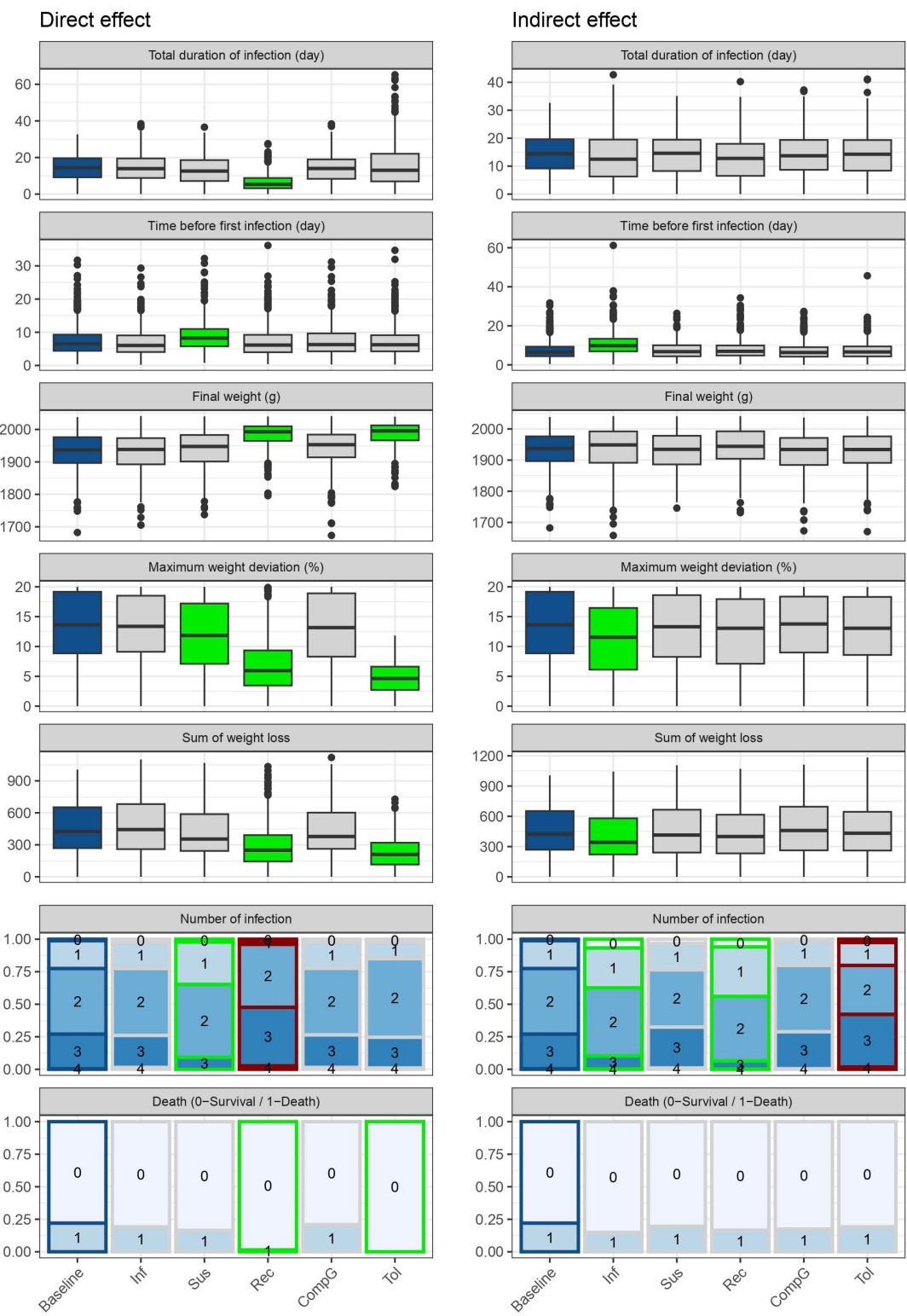

**Fig 8. Comparison of one animal performance between baseline scenario and alternative scenarios.** Inf: decreased infectivity, Sus: decreased susceptibility, Rec: increased recoverability, CompG: increased compensatory growth, Tol: increased tolerance. The direct effect of the trait is calculated

by considering an individual with a favorable trait value (reducing/increasing average trait by one ln unit) among individuals with baseline trait values. The indirect effect is calculated by considering an individual with baseline trait values among individuals with favorable trait values. The color of the boxplots and bar charts corresponds to the associated p-values comparing each scenario to baseline (grey: p>0.05, green: p<0.05 and favorable effect, red: p<0.05 and deleterious effect). Simulations were performed with 500 replicates of 20 birds for each scenario.

Table 6. Comparative impact of host trait improvements on epidemiological and production outcomes. Percentage change relative to baseline for one-unit improvement in each trait. Green: favorable changes; Red: unfavorable changes. Statistical significance from Bonferroni-corrected t-tests: *** p<0.001, ** p<0.01, * p<0.05, ns=not significant.

| | Inf | Sus | Rec | CompG | Tol |
|---|---|---|---|---|---|
| Number of infections | −18.0 *** | −18.0 *** | −9.3 *** | −0.5 ns | +14.6 *** |
| Total duration of infection (days) | −7.7 *** | −6.1 ** | −59.9 *** | −0.9 ns | +23.1 *** |
| Mean prevalence | −12.0 *** | −12.0 *** | −52.0 *** | +0.0 ns | +8.0 *** |
| Death rate (%) | −28.6 *** | −28.6 *** | −95.2 *** | −4.8 ns | −100.0 *** |
| Mean final weight (g) | +0.3 * | +0.2 ns | +3.1 *** | +0.7 *** | +2.5 *** |
| SD of final weight (g) | +6.2 ns | +7.4 * | −42.1 *** | −6.6 ** | −32.1 *** |
| Max infectious load | −68.6 *** | −15.1 *** | −40.0 *** | −1.5 ns | +0.4 ns |
| Mean infectious load | −67.7 *** | −11.7 *** | −50.0 *** | −0.6 ns | +15.4 *** |
| Proportion of final meat loss | −24.0 *** | −20.0 *** | −88.0 *** | −4.0 ns | −88.0 *** |
| Sum of weight loss (g.d) | −7.3 *** | −7.0 *** | −44.7 *** | −5.2 ** | −41.9 *** |
| Area of resilience path | −74.0 *** | −28.9 *** | −73.3 *** | −4.0 ns | −59.0 *** |

Enhanced recoverability (Rec) showed numerous favorable impacts. The direct effects demonstrated dramatically reduced infection duration (−55.58%), improved final weight (+2.77%), reduced maximal weight deviation (−47.13%), reduced sum of weight loss (−37.18%), and nearly eliminated mortality. The indirect effect was also significant for several parameters, with reduced infection duration (−10.54%), improved final weight (+0.60%), reduced maximal weight deviation (−7.69%), reduced sum of weight loss (−7.06%), and reduced mortality (−22.73%). Interestingly, while the direct effect of faster recovery led to an increased number of infections (+19.61%), the indirect effect showed a reduction in the number of infections (−23.04%). The increased number of infections in individuals with enhanced recovery can be attributed to their higher survival rate, allowing them to experience subsequent infections.

Compensatory growth (CompG) exhibited a limited impact, no parameters showing significant change.

Increased tolerance (Tol) showed several strong effects. The direct effect demonstrated highly favorable impacts on weight-related parameters, with improved final weight (+2.90%), dramatically reduced maximal weight deviation (−81.22%), reduced sum of weight loss (−51.04%), and complete elimination of mortality. The indirect effect exhibited no significant impact on weight-related parameters or infection duration, but a significant increase in the number of infections (+8.33%). This pattern highlights how enhanced tolerance, while effectively protecting against production losses, may contribute to increased pathogen circulation in the population by allowing infected animals to maintain high performance levels despite infection.

## Discussion

### Model assumption and limitations

In this study, we applied a transmission model to experimental data to calculate transmission, shedding, oocyst decay, and recovery rates. Among them, transmission rate and recovery rate parameter are both dependent on the number of previous coccidiosis infections, which imply that chickens with different infection histories varied in both susceptibility and

 

infectivity. Specifically, chickens with higher number of previous infections have reduced susceptibility and a lower likelihood of reinfection, and these "experienced" chickens recovered more quickly once infected. To evaluate the model's accuracy, we compared the predicted environmental infection load with observed oocyst counts, finding a good match between the two (Fig 7). We also compared the calculated decay rate with the real-life half-life value, further validating the model's ability to capture the dynamics of infection transmission and decay.

The model was calibrated using literature data on coccidiosis transmission under specific experimental conditions. Several limitations should be noted. First, the data exclusively pertains to *E. acervulina*, and model parameters likely differ for other Eimeria species such as E. tenella or E. maxima, which cause significantly higher mortality rates and more severe pathology. Under conditions of increased pathogen virulence, it may be more important to prioritize host traits that prevent infection or limit transmission over traits that enhance post-infection tolerance. Hence, tolerance-based strategies—which don't directly reduce pathogen transmission—may become substantially less effective. This relates to the well-established concept of trade-offs between virulence and transmissibility in evolutionary epidemiology [36,37]. From the host perspective, tolerance-based strategies—which reduce the fitness cost of infection without directly limiting pathogen replication or transmission—may face analogous trade-offs when virulence is high. Specifically, for tolerance improvements to remain effective under high virulence conditions, the magnitude of improvement would need to be sufficient to prevent mortality and maintain productivity despite severe pathogen loads. Conversely, prevention-focused traits such as lower susceptibility and reduced infectivity may become even more valuable, as they reduce both the probability of infection and the environmental contamination that drives transmission. Similarly, compensatory growth mechanisms would offer diminishing returns if a substantial proportion of infected birds succumb before recovery is possible. However, quantitative predictions with species-specific parameterization of both virulence parameters and epidemiological parameters, which remain poorly characterized in the literature for most Eimeria species, would be required.

Second, the transmission dynamics were based on interactions between experimentally infected and naive birds, while naturally infected birds may exhibit different shedding patterns. Experimental infections typically involve controlled inoculation routes and standardized doses that may not reflect natural exposure pathways and infection dynamics in commercial settings. In particular, experimental inoculations often deliver higher, more synchronized parasite doses than would occur through environmental exposure to sporulated oocysts in litter, potentially leading to more severe and synchronized infection dynamics than observed in field conditions. Additionally, experimentally infected birds may exhibit different shedding kinetics compared to naturally infected birds, as the initial parasite dose and infection route can influence the establishment, proliferation, and subsequent oocyst production patterns. Recognizing these limitations, recent experimental approaches have increasingly used naturally infected "seeder" birds to establish more realistic transmission dynamics. Future model refinements could benefit from parameterization based on such naturalistic transmission experiments, which would better represent the dose-response relationships and temporal dynamics of coccidiosis spread under field conditions.

Third, environmental factors can significantly influence oocyst survival and accumulation dynamics. Oocyst sporulation and decay rates are highly sensitive to temperature and humidity conditions [34], which may vary considerably within and between production cycles. In particular, humidity levels typically increase over time due to feces accumulation and higher housing density, while litter quality deteriorates, potentially creating more favorable conditions for oocyst survival. This creates non-linear pathogen accumulation dynamics that our constant decay rate assumption does not capture: early in the production cycle, high decay rates may limit transmission, while later in the cycle, improved oocyst survival could accelerate environmental contamination. Under such conditions with time-varying decay rates, the indirect benefits of reduced infectivity traits would be amplified, as preventing environmental contamination early would have compounding effects by limiting the pathogen reservoir available during periods of enhanced oocyst survival. Validation under diverse field conditions with monitored environmental parameters would be valuable for quantifying these effects.

Moreover,our model assumes that mortality occurs when birds exceed a 20% weight loss threshold ($G^* = 0.8$), which is admittedly an arbitrary choice due to the absence of empirical data quantifying this relationship for E. acervulina. However, this assumption is biologically justified: severe weight loss in E. acervulina infections indicates chronic malnutrition due to impaired nutrient absorption, intestinal villus damage, and reduced feed intake. Such birds face elevated mortality risk through multiple pathways including starvation, increased susceptibility to secondary bacterial infections, and reduced thermoregulatory capacity. While the exact threshold at which mortality risk becomes substantial is unknown, the existence of a weight loss-mortality relationship is more plausible than its absence for this non-hemorrhagic species. We acknowledge that E. tenella and other hemorrhagic species involve additional mortality mechanisms (acute blood loss, anemia) that are independent of weight loss and would require different modelling approaches. For E. acervulina, an alternative formulation incorporating both weight loss and parasite load and/or host disease resistance as mortality predictors could be explored, but this would introduce additional parameters that cannot be empirically constrained with existing data. Our current formulation represents a parsimonious approach that captures the chronic, nutrition-mediated mortality pathway most relevant to E. acervulina pathogenesis.

Regarding the compensatory growth parameter (c4), we acknowledge limited empirical data for direct calibration in the context of coccidiosis recovery. To assess whether this uncertainty affects our conclusions, we conducted a sensitivity analysis spanning a 100-fold range of compensatory growth capacity (c4 × 0.1 to c4 × 10; see Supplementary Material S2 and S3 Figs). Results demonstrate that epidemic dynamics and trait rankings remain largely stable across this parameter space, as compensatory growth primarily affects post-recovery weight gain rather than infection dynamics themselves. At very high compensatory growth values (c4 × 10), however, improvements in compensatory growth showed greater impact on final weight than improvements in susceptibility or infectivity, suggesting that the relative value of different trait strategies may shift under conditions of exceptionally rapid recovery capacity. Experimental studies tracking individual animal responses to infection, including detailed measurements of growth patterns and mortality risk under various infection intensities, would be valuable for better model calibration.

### Effect of the host traits on individual and group response to coccidiosis

Using the calibrated epidemiological model, we evaluated how different traits related to chicken response to Eimeria affects both individual and group health and performance outcomes. Our results demonstrate that reduced susceptibility had a moderate but significant impact on flock health and performance; reducing average susceptibility by one log unit led to 40% reduction in prevalence and delaying the infection peak by 4 days. This trait primarily operated through direct effects on individual animals, with less susceptible individuals showing reduced infection frequency, improved final weight, and lower mortality. Surprisingly, indirect effects on groupmates were limited, suggesting that the benefits of reduced susceptibility are primarily confined to the individuals carrying the trait rather than providing substantial protection to the overall flock.

Reduced infectivity, modelled as reduced shedding rate, demonstrated substantial impact on disease spread at the flock level, notably decreasing environmental contamination to less than a third of baseline levels and delaying peak infection. As expected, this trait operated primarily through indirect effects, with no significant direct benefits to carriers but substantial benefits to groupmates, including reduced infection numbers, lower mortality, and delayed first infection, highlighting its importance in population-level disease control. Several articles showed a large impact of host infectivity on disease spread [38,39]. Literature [40] showed that super-spreaders, defined as a small proportion of individuals responsible for a disproportionally large number of transmissions, are a common phenomenon in epidemics. In the context of coccidiosis, while substantial variation in oocyst shedding per gram of feces has been documented across individuals [41] the relationship between oocyst shedding and coccidiosis transmission remains unclear. Given that only a few oocysts are sufficient to establish infection in a naive host [42], modest reductions in oocyst shedding may not achieve drastic reduction infection rates, as was the case in our calibrated model as estimates for the transmission coefficient beta

were relatively large. This has been observed for Marek's disease in chicken, where vaccination reduced virus shedding of infected birds but did not block transmission [43,44]. On the other hand, the studies for Marek's disease in chicken revealed significant beneficial effects of reduced virus shedding of infected individuals on disease development and mortality in flock mates, and also reduced virus shedding in these secondary infected birds [44,45]. For chicken coccidiosis, a similar beneficial indirect impact of decreasing oocyst shedding might occur if low shedder birds induced lower symptoms and lower shedding among subsequently infected chicken. Because no data is available to investigate this hypothesis, we did not include such potential indirect beneficial effects in our model.

The present article explored the effect of the host recoverability, defined as the inverse of the infectious period. The beneficial indirect effect occurs due to reduction in the infectious period, which is equal regardless of whether the animal returns to an S state or dies, whereas the direct effect is due to the correlated shorter infected period that impacts on weight loss, but is only beneficial if the animal survives. Enhanced recoverability demonstrated strong benefits at the flock level. A log-unit increase in recovery rate reduced peak prevalence by 25% and significantly improved production parameters including final weight and batch homogeneity since the standard deviation of the weight at culling was decreased 32%. This trait exhibited both strong direct and indirect effects. Directly, it dramatically reduced infection duration and nearly eliminated mortality in carriers with 1 log-unit faster recovery rate compared to the group average (baseline). Indirectly, it benefited groupmates through reduced infection frequency and improved weight parameters, suggesting that faster recovery contributes to both individual and group resilience. The substantial indirect protective effects of infectivity and recoverability highlight the importance of considering both direct and indirect effects of traits when evaluating responses to disease management strategies. This was particularly showing in the comparison of the resilience pathways. The traditional use of resilience pathways aims at measuring individual host resilience by representing the dynamics of health across internal infectious load. Here, we aimed at representing the group level resilience by modelling how the average weight loss of the flock changes with the total infectious load in the environment. We showed that reduced infectivity and increased recoverability had the strongest impact on the loop area, by simultaneously limiting infectious load in the environment and maintaining target growth.

Increasing tolerance by one log unit showed remarkable impact on flock performance, reducing weight deviation to just 4% despite higher infection rates, and significantly improving final weight. The effects showed an interesting direct/indirect trade-off: directly, it eliminated mortality and dramatically improved weight parameters in carriers, but led to increased infection duration. Indirectly, it resulted in higher infection frequency in groupmates, highlighting potential negative epidemiological consequences of enhanced tolerance. This phenomenon underscores the potential risks of vaccines that mask symptoms without preventing transmission, particularly in scenarios where population vaccination is incomplete or where some individuals fail to respond to vaccination.

At the flock level, enhanced compensatory growth showed limited impact, with patterns similar to baseline except for slightly faster recovery of weight post-infection. The effects were predominantly direct, with carriers showing improved final weight and a tendency for reduced total weight loss. Indirect effects were negligible, suggesting that this trait operates primarily at the individual level without significantly affecting group dynamics. This limited impact may be due to the baseline compensatory growth rate being sufficiently rapid to allow almost complete recovery by 84 days of age, following infection at day 5. Therefore, increasing the rate of compensatory growth did not substantially alter outcomes. Additionally, while compensatory growth rate likely affects mean feed efficiency over the flock's lifetime, this aspect was not included in our model. More broadly, compensatory growth is a key component of animal resilience to disease. Resilience refers to animal's capacity to 'bounce back' after a relatively short-term disturbance [46]. This capacity directly influences how well animals can handle successive challenges. Our simplified model considered only one type of perturbation (coccidiosis infection), and although rapid compensatory growth could reduce maximum growth loss during subsequent infections, it did not affect susceptibility to future Eimeria infections or other pathogens. While our model did not demonstrate a strong effect of compensatory growth on overall production, disease resilience in its broader sense likely plays a more significant

role in flock health during coccidiosis epidemics, particularly when considering multiple environmental challenges and their interactions.

## Implications

Our modeling framework could help better estimate the effects of different coccidiosis control methods on flock health by predicting how interventions targeting specific host traits will impact both individual and group-level outcomes. The effectiveness of vaccination strategies will depend on both the vaccine type and the targeted host traits. Traditional live vaccines may primarily enhance recoverability and reduce susceptibility by stimulating natural immunity, while also potentially decreasing infectivity through reduced oocyst shedding [47–49]. However, the case of leaky vaccines presents a particular challenge – these vaccines increase host tolerance but do not prevent infection or transmission, potentially selecting for more virulent pathogen strains. Anticoccidial medications in chicken feed work by either inhibiting specific stages of the coccidia life cycle (coccidiostats) or by killing the parasites directly (coccidiocides). By disrupting oocyst development and reducing parasite multiplication within intestinal cells, these drugs primarily impact host susceptibility by preventing initial infection establishment, enhance recoverability by reducing parasite burden, and decrease infectivity by limiting oocyst production and environmental contamination [50]. The gut microbiota represents a promising intervention target that could modify all host trait responses to coccidiosis [50]. This approach is particularly relevant for enteric pathogens like coccidia, as the intestinal microbiome could influence pathogen colonization, immune responses, and gut barrier function. Microbiota composition can be strategically modified through dietary interventions, feed additives, probiotics, or postbiotics to enhance natural resistance mechanisms [51,52]. Moreover, strategic removal of high-shedding individuals could theoretically decrease mean flock infectivity and reduce environmental pathogen burden [40]. However, this approach is easier when superspreaders can be easily identified through visible clinical signs. Unfortunately, for coccidiosis, the animals with the highest infectivity (oocyst shedding) are not necessarily those displaying the most severe clinical symptoms, making identification and targeted culling challenging to implement effectively [41].

The growing resistance of Eimeria species to coccidiostatic drugs [53,54] and the high cost of vaccines have emphasized the need for alternative intervention strategies, with genetic improvement emerging as a promising approach. Our approach could be used to simulate the effect of a genetic selection on a given trait by incorporating an environmental and a genetic component to the five traits investigated. Selection for infectious disease resistance or resilience traits requires a deep knowledge of the epidemiology of the disease in order to understand how improvements in individual traits affect the disease dynamics in the population [3,55,56]. Disease epidemiology is rarely incorporated into genetic selection response assessments, which can lead to underestimation of the true genetic variance underlying disease transmission [56,57]. This oversight may result in missing significant opportunities for genetic improvement in disease resistance [55].

Inter-individual variability in host response to coccidiosis has been documented across several key traits, highlighting the potential for genetic improvement. Multiple studies have highlighted the potential for genetic selection targeting chicken tolerance to coccidiosis. Remarkably, one article [58] documented significant differences in growth depression among chicken breeds infected with *E. tenella*, with Fayoumi males exhibiting only 9.2% growth reduction compared to 20–28% in other breeds. While specific estimates for genetic variability in host susceptibility, infectivity and recoverability from Eimeria infection in chickens is limited, studies in other species suggest that substantial genetic variation in these traits exists [59–61].

Until recently, estimating genetic variation in these traits was hampered by a lack of models that can provide such estimates from disease data. However, recently, Pooley et al. [4,62] introduced a new Bayesian methodology and software tool that estimates gene effects and polygenic contributions to host susceptibility, infectivity, and recoverability from epidemic data. These may be applicable to avian coccidiosis, should appropriate data exist. However, the actual potential for genetic improvement through selection depends on both phenotypic and genetic variance of these traits, which remain unknown. Higher phenotypic and genetic variance for a particular trait could enhance its potential for genetic selection,

even if its comparative impact in our standardized analysis was lower. Therefore, expansion of the model introduced in this paper to a genetic-epidemiological model and estimation of genetic parameters for these traits would be crucial for fully evaluating their potential in breeding programs. In a genetic-epidemiological modelling study, Gharbi et al. [63] showed that after 10 generations of selection on the best 0.01 of populations, treatments may become unnecessary to control sea lice populations in Atlantic salmon farms, while six treatments would be required in unselected populations to maintain lice numbers below acceptable thresholds.

Moreover, limited knowledge exists regarding genetic correlations among susceptibility, infectivity, recoverability, and tolerance for coccidiosis. These traits may share underlying physiological or immunological mechanisms. For instance, animals with lower susceptibility might also exhibit enhanced recoverability due to a superior ability to suppress pathogen multiplication in the gut. Furthermore, documented trade-offs between growth and immunity in the literature suggest potential adverse effects of selecting for health traits [64–66]. However, [67] demonstrated that genetic selection for bovine tuberculosis resistance would have limited impact on other breeding traits since genetic correlations were mostly non-significant, while favorably correlating with the overall profitability index, suggesting that selection for disease resistance could be incorporated into breeding programs without negative consequences on other production traits. Understanding these genetic relationships and trade-offs is essential for developing effective breeding strategies that balance health and production traits.

Finally, advancing our understanding of coccidiosis epidemiology and host trait effects requires systematic collection and sharing of detailed field data through coordinated initiatives. Current knowledge gaps reflect the scarcity of commercial farm data with individual-level resolution (longitudinal growth, infection status, oocyst shedding dynamics). One should incentivize standardized data collection protocols that capture individual bird trajectories rather than only flock-level summaries, as such data are essential for parameterizing and validating epidemiological models [68].

## Conclusion

Using an epidemiological model that we calibrated for coccidiosis in poultry based on published literature data, we evaluated how diverse disease-resilience traits affect both individual and group health and performance outcomes. Our analysis revealed that in order to reduce environmental pathogen burden, infectivity emerged as the key target trait. For minimizing impacts on body weight, tolerance was most important, though this came with some negative effects on pathogen load and prevalence. Recoverability – the animal's capacity to rapidly eliminate the pathogen from its system – was the trait that achieved both objectives, maximizing flock resilience to coccidiosis. Additionally, our study examined both direct effects (on the infected individual) and indirect effects (on flock-mates through reduced environmental contamination). It showed the strong part of indirect effect for infectivity and recoverability traits, highlighting the importance to take it into account when evaluating a disease control strategy.

## Supporting information

**S1 Table. States of Each Pair in Experiment I: 5-Oocyst Group (9 pairs).**
(DOCX)

**S2 Table. States of Each Pair in Experiment II: 50-Oocyst Group (17 pairs).**
(DOCX)

**S3 Table. States of Each Pair in Experiment III: 500-Oocyst Group (9 pairs). Bird 525C had only 2 out of 30 records. Due to its lackness, the pair (of 546I and 525C) is excluded from the analysis).**
(DOCX)

**S4 Table. States of Each Pair in Experiment IV: 5000-Oocyst Group.**
(DOCX)

**S5 Table. Comparison of flock performance between baseline scenario (all traits value = 0) and scenarios where the average individual trait value was set to 1 (or −1 for infectivity and susceptibility). Outcomes are presented as mean (standard deviation) with associated p-values from t-tests comparing each scenario to baseline. Simulations were performed with 500 replicates of 20 birds for each scenario.**
(DOCX)

**S6 Table. Comparison of one animal performance between baseline scenario and alternative scenarios (F: decreased infectivity, G: decreased susceptibility, M: increased recoverability, R: increased compensatory growth, K: increased tolerance).** The direct effect of the trait is estimated by considering an individual with a favorable trait among individuals with baseline traits. The indirect effect is estimated by considering an individual with baseline trait among individual with favorable traits. Values are presented as mean (standard deviation), with p-values comparing each scenario with baseline. Simulations were performed with 500 replicates of 20 birds for each scenario.
(DOCX)

**S1 Fig. Profile-Likelihood Plots for Decay ($\lambda$), Shedding ($\sigma$), Transmission ($\beta$ and $a_t$ in $\frac{\beta}{a_\beta^{ninf}}$) Rate Parameters.** The x-axis shows parameter values, and the y-axis shows AIC values. The solid line represents profile likelihood AIC values, with the lowest point (red star) indicating the best estimate. Black dots mark critical values for the 95% confidence interval, defined as the minimum AIC plus 2 (dashed line).
(PNG)

**S2 Fig. To assess the robustness of our conclusions to uncertainty in the compensatory growth parameter (c4), we conducted a sensitivity analysis testing five scenarios spanning a 100-fold range: c4 × 0.1, c4 × 0.25, c4 baseline, c4 × 4, and c4 × 10.** For each c4 value, we simulated all six trait scenarios (baseline, lower susceptibility, lower infectivity, higher recoverability, improved compensatory growth, and improved tolerance) with 20 replications each, yielding 600 simulations total. The figure shows that epidemic dynamics (number of infected individuals, infectious load, mean weight trajectories) remain nearly identical across all c4 values. Weight deviation patterns scale predictably with c4: higher values produce faster post-recovery weight gain (steeper decline in weight deviation after peak infection), but maximum weight deviation during acute infection remains similar across scenarios. Importantly, trait rankings remain stable across the entire parameter space for epidemic-related outcomes.
(PNG)

**S3 Fig. Presents comparative boxplots of key outcome variables (final weight, sum of weight loss, infection metrics) for each trait improvement strategy across all c4 values in the sensitivity analysis presented in supplementary figure S2.** The relative performance of different trait improvement strategies shows consistent patterns across most of the c4 range. However, at the highest compensatory growth value tested (c4 × 10), improvements in compensatory growth show enhanced benefits on final weight and cumulative weight loss relative to susceptibility and infectivity improvements, indicating that the relative value of trait improvement strategies may shift under conditions of exceptionally rapid compensatory growth capacity.
(PNG)

## Acknowledgments

We gratefully acknowledge Dr. Jamie Prentice for his valuable assistance throughout this research project.

## Author contributions

**Conceptualization:** Marie ITHURBIDE, Marie-Hélène Pinard van der Laan.

**Formal analysis:** Marie ITHURBIDE, Yuqi Gao.

**Funding acquisition:** Marie-Hélène Pinard van der Laan.

**Investigation:** Marie ITHURBIDE, Andries D. Hulst.

**Methodology:** Marie ITHURBIDE, Yuqi Gao, Andrea Doeschl-Wilson.

**Supervision:** Marie-Hélène Pinard van der Laan, Mart C.M. De Jong, Andrea Doeschl-Wilson.

**Writing – original draft:** Marie ITHURBIDE, Yuqi Gao.

**Writing – review & editing:** Marie-Hélène Pinard van der Laan, Yuqi Gao, Andries D. Hulst, Mart C.M. De Jong, Andrea Doeschl-Wilson.

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
