## [Decision Letter · Decision Letter 0]

13 Nov 2025

Dear Dr. ITHURBIDE,

Thank you for submitting your manuscript to PLOS ONE. After careful consideration, we feel that it has merit but does not fully meet PLOS ONE’s publication criteria as it currently stands. Therefore, we invite you to submit a revised version of the manuscript that addresses the points raised during the review process.

We look forward to receiving your revised manuscript.

Kind regards,

Shawky M Aboelhadid, PhD

Academic Editor

PLOS ONE

**Journal Requirements:**

”European Union's Horizon Europe Project 10113646 EUPAHW.”

“This work was co-funded by the European Union's Horizon Europe Project 10113646 745 EUPAHW.”

”European Union's Horizon Europe Project 10113646 EUPAHW.”

5. We note you have included a table to which you do not refer in the text of your manuscript. Please ensure that you refer to Table 2 in your text; if accepted, production will need this reference to link the reader to the Table.

**Additional Editor Comments:**

The authors needs to revise the manuscript in accordance to the comments of the reviewers, with special attention to add more details in the section on materials and methods

Reviewers' comments:

Reviewer's Responses to Questions

**Comments to the Author**

1. Is the manuscript technically sound, and do the data support the conclusions?

Reviewer #1: Yes

2. Has the statistical analysis been performed appropriately and rigorously?

Reviewer #1: Yes

3. Have the authors made all data underlying the findings in their manuscript fully available?

Reviewer #1: Yes

4. Is the manuscript presented in an intelligible fashion and written in standard English?

Reviewer #1: Yes

Reviewer #1: The model proposed by the authors appears intriguing and would contribute significantly to the treatment of avian coccidiosis. However, the work contains some errors that would need to be corrected to improve the manuscript. It is important for the authors to remember that every time the scientific name of a species (genus or species) is mentioned, it must be written in italics, in accordance with international standards for taxonomic nomenclature.

The manuscript often mixes the figure legends with the informative text. For example, in the introduction section, the paragraph between lines 53 and 67 should not be there; the figure legends should be at the end of the manuscript, after the reference list.

Some information in the introduction lacks bibliographic support (e.g., line 73: ...and 5- compensatory growth occurring after the infection ends.)

Some paragraphs are excessively long, which makes it difficult to understand the message.

Some citations do not conform to the journal's style (e.g., line 180: Voeten, 1989; line 197: Velkers (2010), et al.). Please review and correct this.

**Do you want your identity to be public for this peer review?** For information about this choice, including consent withdrawal, please see our Privacy Policy

Reviewer #1: No

---

## [Author Response · Author response to Decision Letter 1]

4 Dec 2025

We thank the editor and reviewer for their thorough review and constructive comments, which have significantly improved the quality and clarity of our manuscript. We have carefully addressed all comments point-by-point below and made the corresponding changes throughout the revised manuscript. We believe these revisions have strengthened our work and we are grateful for the valuable feedback.

Review Comments to the Author

Reviewer #1: The model proposed by the authors appears intriguing and would contribute significantly to the treatment of avian coccidiosis. However, the work contains some errors that would need to be corrected to improve the manuscript. It is important for the authors to remember that every time the scientific name of a species (genus or species) is mentioned, it must be written in italics, in accordance with international standards for taxonomic nomenclature.

Response: Corrected. All scientific names (genus and species) are now properly italicized in accordance with international standards for taxonomic nomenclature. We reviewed every instance of species names including Eimeria spp. and other organisms mentioned in the text.

The manuscript often mixes the figure legends with the informative text. For example, in the introduction section, the paragraph between lines 53 and 67 should not be there; the figure legends should be at the end of the manuscript, after the reference list.

Response: Corrected. All figure legends have been moved to the end of the manuscript, after the References section, as per PLOS ONE requirements.

Some information in the introduction lacks bibliographic support (e.g., line 73: ...and 5- compensatory growth occurring after the infection ends.)

Response: We thank the reviewer for this helpful observation. We agree that several statements in the introduction required additional bibliographic support. We have now added several references throughout the introduction to address this issue. Specifically, we have incorporated references 3 and 4 (Doeschl-Wilson et al., 2021; Pooley et al., 2020) to support our statements on disease resilience and individual variation in infection dynamics. We have also added references 7, 8, 11, 12, and 13 (Henken et al., 1994; Martin and Sauvant, 2010; Yun et al., 2000; Voeten et al., 1988) to better document the mechanisms of coccidiosis infection, immune responses, and compensatory growth effects on broiler performance. These additions provide a more comprehensive and well-supported framework for our study.

Some paragraphs are excessively long, which makes it difficult to understand the message.

Response: Corrected. We revised the manuscript to improve readability by breaking up long paragraphs.

Some citations do not conform to the journal's style (e.g., line 180: Voeten, 1989; line 197: Velkers (2010), et al.). Please review and correct this.

Response: Corrected. We have reviewed and corrected all citations to conform to PLOS ONE style.

---

## [Decision Letter · Decision Letter 1]

23 Dec 2025

Dear Dr. ITHURBIDE,

Thank you for submitting your manuscript to PLOS ONE. After careful consideration, we feel that it has merit but does not fully meet PLOS ONE’s publication criteria as it currently stands. Therefore, we invite you to submit a revised version of the manuscript that addresses the points raised during the review process.

We look forward to receiving your revised manuscript.

Kind regards,

Sanaullah Sajid, M.Phil/PhD

Academic Editor

PLOS One

Journal Requirements:

**Additional Editor Comments:**

Reviewer 1 forget to attach his suggestion. I am sharing the suggestions here

Suggestions

The manuscript is technically sophisticated and addresses a critical gap in infectious disease modeling for livestock. However, the conclusions regarding compensatory growth are weakened by the use of arbitrary parameter values and a lack of sensitivity analysis for these specific variables.

Major Weaknesses and Critical Issues

1. Parameters for compensatory growth (p) and mortality rates (µ) were set arbitrarily due to a lack of specific literature data. Because these traits are core to the paper’s conclusions regarding resilience, the results for scenario “CompG” and “Tol” may be artifacts of these specific values rather than generalizable biological truths.

2. The model is calibrated exclusively on E. acervulina data. Other species like E. tenella or E. maxima cause significantly higher mortality and different growth depression patterns. The conclusions may not hold for the most economically damaging forms of coccidiosis. Fix by adding a discussion/simulation of “high-virulence” parameters.

3. The model assumes a constant decay rate (λ) for oocysts, whereas oocyst survival is highly dependent on environmental humidity and temperature. In real-world poultry houses, pathogen accumulation is non-linear and affected by litter quality, which could drastically change the indirect benefits of lower infectivity.

4. In Table 1 and Equation [iv], the authors define the infection probability. Please clarify if N is the total population or the number of susceptible individuals, as this affects the density-dependence vs. frequency-dependence assumptions of the model.

5. The mortality trigger G* is set at 0.8 (20% weight loss). While Table 4 shows deviations up to 28.9%, mortality in coccidiosis is often due to acute intestinal hemorrhage (for E. tenella) rather than just weight loss. The authors should justify why weight loss is a sufficient proxy for mortality across different infection intensities.

6. On lines 164-168, growth is modeled proportional to the difference between current and theoretical weight. I suggest the authors test if a leaky recovery model, where some growth potential is permanently lost would change the resilience loop area significantly.

7. In Fig 5, the units of L(t) are scaled and difficult to compare directly with oocyst counts. A dual-axis or normalized plot would be more intuitive.

8. Coccidiosis is a parasitic disease... caused by... Eimeria. Since this is mentioned in the abstract, the opening of the introduction is slightly repetitive.

9. Assumed exposed period of 4 days. Provide a brief range if available for other species.

10. Parameter sigma (Shedding rate) is listed as 2.84 d-1. Please ensure the units are consistent with oocysts per unit time or infectious load units.

Reviewer Suggestion on Requested References:

As a reviewer, I have examined additional references suggested for inclusion.

In the Discussion (Lines 1053–1056), where you discuss how vaccines enhance recoverability and reduce susceptibility, the following works provide excellent comparative context for poultry immunoprophylaxis:

Sajid, S., ur Rahman, S., & Mohin, M. (2024). Development of egg yolk-based polyclonal antibodies and immunoprophylactic potential of antigen-antibody complex against infectious bursal disease. Veterinary and Animal Science, 23, 100326.

Sajid, S., ur Rahman, S., & Mohsin, M. (2022). IgY: a key isotype and promising antibody for the immunoprophylaxis therapy of infectious bursal disease virus infections. Microbiology and Biotechnology Letters, 50(3), 430-435.

Sajid, S., & Mohsin, M. (2022). Development of an Immune Complex Vaccine against Infectious Bursal Disease Virus and its Potential Response in Poultry Birds. Iranian Journal of Medical Microbiology, 16(6), 528-536.

Introduction (Line 249), you mention nutrition as a factor influencing host traits. This study on turnip powder in cereal blends could support the nutritional intervention aspect:

Toor, I. F., Sajid, S., Akmal, A., Abidin, Z. U., Fatima, Z., Althawab, S. A., ... & Alsulami, T. (2025). Nutritional Evaluation of Turnip Powder in Cereal Blends: A Study on Wheat, Oats, and Turnips. Food Science & Nutrition, 13(5), e70157.

In the Discussion (Lines 1060–1063), you mention gut microbiota as a target for modifying host responses. This work on Lactobacillus could support that specific point:

Farzand, I., ur Rahman, S., Sajid, S., & Nayab, S. (2020). Evaluation of modified MRS media for the selective enumeration of Lactobacillus casei. Pure and Applied Biology. Vol. 10, Issue 1, pp194-198.

The works on Infectious Bursal Disease Virus (IBDV) and Mycoplasma demonstrate the variability of host responses in Pakistani broiler populations. These could be used in the Discussion when you talk about the need for local epidemiological data.

Sajid, S., Rahman, S. U., Mohsin Gilani, M., Sindhu, Z. U. D., Ali, M. B., Hedfi, A., ... & Mahmood, S. (2021). Molecular characterization and demographic study on infectious bursal disease virus in faisalabad district. Plos one, 16(8), e0254605.

The following papers could strengthen your Discussion regarding alternative intervention strategies:

Chen, J., Chen, F., Peng, S., Ou, Y., He, B., Li, Y., & Lin, Q. (2022). Effects of Artemisia argyi Powder on Egg Quality, Antioxidant Capacity, and Intestinal Development of Roman Laying Hens. Frontiers in physiology, 13, 902568. https://doi.org/10.3389/fphys.2022.902568

Hassan, F. U., Liu, C., Mehboob, M., Bilal, R. M., Arain, M. A., Siddique, F., Chen, F., Li, Y., Zhang, J., Shi, P., Lv, B., & Lin, Q. (2023). Potential of dietary hemp and cannabinoids to modulate immune response to enhance health and performance in animals: opportunities and challenges. Frontiers in immunology, 14, 1285052. https://doi.org/10.3389/fimmu.2023.1285052

Liu, H., Bing, P., Zhang, M., Tian, G., Ma, J., Li, H., Bao, M., He, K., He, J., He, B., & Yang, J. (2023). MNNMDA: Predicting human microbe-disease association via a method to minimize matrix nuclear norm. Computational and structural biotechnology journal, 21, 1414–1423. https://doi.org/10.1016/j.csbj.2022.12.053

Reviewers' comments:

Reviewer's Responses to Questions

**Comments to the Author**

Reviewer #1: All comments have been addressed

Reviewer #2: (No Response)

2. Is the manuscript technically sound, and do the data support the conclusions?

Reviewer #1: Yes

Reviewer #2: Yes

3. Has the statistical analysis been performed appropriately and rigorously?

Reviewer #1: Yes

Reviewer #2: Yes

4. Have the authors made all data underlying the findings in their manuscript fully available?

Reviewer #1: Yes

Reviewer #2: Yes

5. Is the manuscript presented in an intelligible fashion and written in standard English?

Reviewer #1: Yes

Reviewer #2: Yes

Reviewer #1: I congratulate the authors on their commendable work in enhancing the quality of their manuscript...

Reviewer #2: The manuscript is technically sophisticated and addresses a critical gap in infectious disease modeling for livestock. However, the conclusions regarding compensatory growth are weakened by the use of arbitrary parameter values and a lack of sensitivity analysis for these specific variables.

**Do you want your identity to be public for this peer review?** For information about this choice, including consent withdrawal, please see our Privacy Policy

Reviewer #1: No

Reviewer #2: No

---

## [Author Response · Author response to Decision Letter 2]

9 Feb 2026

RESPONSE TO REVIEWERS - ROUND 2

We sincerely thank the editor and both reviewers for their thorough evaluation and constructive feedback. We particularly appreciate Reviewer #2's detailed technical comments, which have helped us strengthen the manuscript substantially. We have carefully addressed all comments point-by-point below and made the corresponding changes throughout the revised manuscript.

REVIEWER #1

Comment: I congratulate the authors on their commendable work in enhancing the quality of their manuscript... All comments have been addressed.

Response: We thank Reviewer #1 for their positive evaluation and for confirming that our previous revisions adequately addressed all their concerns.

REVIEWER #2

We thank Reviewer #2 for their comprehensive and insightful review. Their technical expertise has helped us identify important areas for clarification and improvement. Below, we address each comment in detail.

Major Comment 1: Arbitrary parameter values for compensatory growth and mortality

Comment: Parameters for compensatory growth (p) and mortality rates (µ) were set arbitrarily due to a lack of specific literature data. Because these traits are core to the paper's conclusions regarding resilience, the results for scenario "CompG" and "Tol" may be artifacts of these specific values rather than generalizable biological truths.

Response: We thank the reviewer for this important concern regarding parameter justification and model robustness.

Regarding tolerance parameters: We respectfully disagree that these were set arbitrarily. Our tolerance parameterization is based on empirical data from the studies cited in Table 4, which document weight losses ranging from 10% to 28.9% across different Eimeria species and chicken breeds. Our model reproduces weight deviations in this biologically realistic range (see "Determination of growth parameters" section), confirming that our tolerance parameters are appropriately calibrated to observed infection impacts. The variation in Table 4 specifically illustrates inter-breed differences in tolerance—a key biological reality that motivates our trait-based modeling approach.

Regarding compensatory growth parameters: We acknowledge that the compensatory growth parameter (c4) has less direct empirical support, as no studies have quantified compensatory growth rates following coccidiosis specifically. To address this concern and assess the robustness of our conclusions, we conducted a comprehensive sensitivity analysis.

Sensitivity analysis design: We tested five scenarios spanning a 100-fold range of compensatory growth capacity: c4 × 0.1, c4 × 0.25, c4 baseline, c4 × 4, c4 × 10.

For each c4 value, we simulated all six trait scenarios (baseline, lower susceptibility, lower infectivity, higher recoverability, compensatory growth, tolerance) with 20 replications each (600 simulations total).

Results demonstrate that epidemic dynamics are minimally affected by c4 variation (Figure 1): The number of infected individuals, infectious load, and mean weight trajectories show nearly identical patterns across all c4 values. Compensatory growth primarily affects post-recovery weight gain rather than infection dynamics themselves.

Weight deviation patterns scale predictably with c4 : Higher c4 values produce faster recovery (steeper decline in weight deviation after peak infection), but the maximum weight deviation during acute infection remains similar across scenarios (Figures 1 and 2). Hence, trait rankings that referred to epidemic dynamic remain stable across the entire parameter space (Figure 2).

However, for very high compensatory growth value (c4 x 10), an improved compensatory growth has more impact on final weight (+1.7% for increased compensatory growth vs baseline in c4x10 scenario vs +1.5% in the c4 article value) and sum of weight loss (-38% for increased compensatory growth vs baseline in c4x10 scenario vs -22% in the c4 article value) than an improved susceptibility and infectivity.

We added a paragraph in the discussion section about the uncertainty in c4 and the sensitivity analysis (L598-607). We added the figure 1 and 2 in supplementary materials.

Figure 1 : Impact of compensatory growth parameter value on infection and body weight dynamics

Figure 2 : Relative Impact of compensatory growth parameter value on body weight deviations

Major Comment 2: Model calibrated only on E. acervulina data

Comment: The model is calibrated exclusively on E. acervulina data. Other species like E. tenella or E. maxima cause significantly higher mortality and different growth depression patterns. The conclusions may not hold for the most economically damaging forms of coccidiosis. Fix by adding a discussion/simulation of "high-virulence" parameters.

Response: We appreciate this important observation. However, we lack sufficient literature data to properly parameterize the epidemiological components (transmission rate β, shedding rate σ, decay rate λ) for other Eimeria species. While mortality and growth depression differ among species, transmission parameters would also differ substantially due to species-specific differences in tissue tropism, oocyst production, and within-host dynamics. Without empirical data to constrain these parameters, we prefer not to present speculative simulations.

Instead, we have added a discussion section (lines 582-597) addressing how increased virulence would modify the relative importance of host traits. With more virulent species, moderate improvements in prevention-focused traits (lower susceptibility, lower infectivity) may become even more critical, while moderate improvements in tolerance would become less impacting on death rate and weight loss.

In order to support our point, we performed exploratory simulations (not included in the manuscript) reducing the mortality threshold G* from 0.8 to 0.9 (10% vs. 20% weight loss). Results (Figure 3) show that: (1) negative effects of tolerance are amplified when mortality risk is higher, and (2) positive effects of reduced infectivity are enhanced when infections are more lethal.

Figure 3. Impact of host traits on disease dynamics and flock performance in alternative high mortality model. Comparison between baseline scenario and scenarios with modified traits: reduced infectivity (Inf), reduced susceptibility (Sus), increased recoverability (Rec), increased compensatory growth (CompG) and increased tolerance (Tol). For each scenario the corresponding parameter was increased or decreased of 1 ln unit compared with baseline.

Major Comment 3: Constant oocyst decay rate assumption

Comment: The model assumes a constant decay rate (λ) for oocysts, whereas oocyst survival is highly dependent on environmental humidity and temperature. In real-world poultry houses, pathogen accumulation is non-linear and affected by litter quality, which could drastically change the indirect benefits of lower infectivity.

Response: We thank the reviewer for highlighting this simplification. We acknowledge that environmental variation in oocyst survival is an important factor in real-world settings. We have now:

• Added explicit description of this assumption in the Methods section (lines 153-155), noting that our constant decay rate represents an average across typical broiler house conditions.

• Included in the Discussion (lines 570-581) a paragraph addressing how environmental heterogeneity in oocyst survival might modify our results

We agree that incorporating environmental heterogeneity would be valuable for farm-specific predictions, though the core biological insights regarding hosts traits remain valid under the constant-decay simplification.

Major Comment 4: Clarification of N in infection probability

Comment: In Table 1 and Equation [iv], the authors define the infection probability. Please clarify if N is the total population or the number of susceptible individuals, as this affects the density-dependence vs. frequency-dependence assumptions of the model.

Response: We thank the reviewer for this comment, though we are somewhat surprised by it as the variable N does not appear in Table 1 or in our infection probability formulation. In our model, the infection probability is proportional to A, which represents the area (m²) in which birds are housed. This formulation inherently corresponds to a density-dependent transmission model (equivalent to holding N constant while varying contact rates with environmental contamination). Birds housed at higher densities (same N, smaller A) have higher per-capita infection rates due to increased contact with contaminated litter.

To clarify this point, we have now added explicit text in the Materials and Methods section (lines 126-129) stating: "Our transmission formulation is density-dependent: infection probability scales with pathogen load per unit area (L(t)/A), meaning that higher stocking densities (more birds per m²) result in higher per-capita infection rates. This is consistent with field observations showing that coccidiosis prevalence increases with stocking density in commercial production systems."

This density-dependent formulation is biologically appropriate for coccidiosis transmission in housed poultry, where contact with contaminated litter (rather than direct bird-to-bird contact) is the primary transmission route, and contact rates with contaminated surfaces increase with stocking density.

Major Comment 5: Weight loss as mortality proxy

Comment: The mortality trigger G* is set at 0.8 (20% weight loss). While Table 4 shows deviations up to 28.9%, mortality in coccidiosis is often due to acute intestinal hemorrhage (for E. tenella) rather than just weight loss. The authors should justify why weight loss is a sufficient proxy for mortality across different infection intensities.

Response: We thank the reviewer for this comment, which allows us to clarify an important aspect of our parameterization approach. Table 4 includes data from multiple Eimeria species, not exclusively E. acervulina. Indeed, studies using E. tenella show greater weight losses (such as the 28.9% mentioned) and involve different pathogenic mechanisms—E. tenella causes hemorrhagic cecal lesions where mortality is more directly linked to blood loss and acute intestinal damage rather than chronic weight loss. The pathogenesis of E. tenella is fundamentally different from E. acervulina.

We deliberately chose to include studies on other Eimeria species in Table 4 to illustrate the range of tolerance levels across chicken breeds (particularly the studies by Hamet and Pinard, which worked exclusively with E. tenella). These studies highlight a central point of our article: the substantial inter-individual and inter-breed variability in disease response, which is the biological basis for the host trait variation we model. This variability in tolerance is relevant regardless of the specific Eimeria species involved.

For E. acervulina specifically, no published study clearly establishes the quantitative relationship between weight loss and mortality, and our 20% threshold (G = 0.8) is admittedly set arbitrarily. However, there are biological reasons to expect a relationship between weight loss and mortality for E. acervulina: severe weight loss indicates chronic malnutrition, compromised nutrient absorption due to intestinal damage, reduced competitive ability at feeders, and increased susceptibility to secondary bacterial infections—all of which elevate mortality risk. While the exact threshold is uncertain, it is more biologically plausible that such a relationship exists than that it does not exist. An alternative approach would have been to make mortality depend on both weight loss and another variable (e.g., parasite load or host disease resistance), but in the absence of empirical data, this would have introduced additional arbitrary parameter choices without improving model realism.

We have added a paragraph in the Discussion (lines 516-533) addressing this limitation and its implications.________________________________________

Major Comment 6: Testing leaky recovery model

Comment: On lines 164-168, growth is modeled proportional to the difference between current and theoretical weight. I suggest the authors test if a leaky recovery model, where some growth potential is permanently lost, would change the resilience loop area significantly.

Response:

This is an insightful suggestion. We implemented and tested a leaky recovery variant by introducing a dynamic weight_potential variable for each individual. In the original model, this potential increases at a constant rate and remains unaffected by infection. In the leaky recovery variant, during infection, the growth potential decreases proportionally to the magnitude of weight deficit: the larger the gap between potential weight and actual weight (i.e., the more severe the growth depression), the more growth potential is permanently lost. This captures the biological reality that severe intestinal damage during critical growth periods may permanently impair nutrient absorption capacity and growth trajectory.

Results (Figure 4) show that the leaky recovery mechanism has minimal impact on epidemic dynamics (50 simulations with 20 individuals each): both the number of infected animals and environmental infectious load follow nearly identical trajectories in leaky versus non-leaky models. The weight deviation remains higher in the leaky model in the late state of the epidemic, confirming that some growth potential is irreversibly lost.

Despite these limited differences, the relative ranking of host traits and our core conclusions remain qualitatively unchanged. We have therefore chosen not to incorporate this additional complexity into the main manuscript, as it would require extensive additional parameterization without fundamentally altering our findings.

Figure 4: Comparison of model results between a ‘leaky recovery model’ and a model with invariant growth potential

Major Comment 7: Figure 5 - L(t) units and scaling

Comment: In Fig 5, the units of L(t) are scaled and difficult to compare directly with oocyst counts. A dual-axis or normalized plot would be more intuitive.

Response: We thank the reviewer for this comment, which allows us to clarify an important conceptual aspect of our model. In our framework, L(t) represents infectious load rather than a direct oocyst count. This is a deliberate modeling choice for the following reasons:

• Infectious load is not linearly related to oocyst counts. The relationship between the number of oocysts shed and their actual infectivitious load depends on multiple factors including oocyst sporulation status, environmental conditions affecting sporulation rates, faces excretion faces behaviour / consistency / quantity. A single "infectious unit" in our model represents the effective transmission potential, which may correspond to different numbers of physical oocysts under different conditions.

• L(t) is a virtual epidemiological quantity that captures transmission potential rather than a directly measurable physical quantity. This approach is standard in epidemiological modeling where the "force of infection" may integrate multiple unmeasured or variable factors. Converting L(t) to oocyst counts would require assumptions about sporulation rates, oocyst viability, and dose-response relationships that would introduce additional uncertainty without improving the model's predictive or comparative value.

To improve clarity, we have now:

• Enhanced the figure caption to explicitly state: "(L(t), a dimensionless measure of transmission potential)”

• Added clarifying text in the Methods section (lines 233-236) explaining the concept of infectious load and why it cannot be directly converted to oocyst counts: "We model infectious load L(t) as a virtual epidemiological quantity representing effective transmission potential. This cannot be directly converted to oocyst counts because the relationship between oocyst numbers and infectivity is non-line

---

## [Editor Report · Decision Letter 2]

11 Feb 2026

Understanding the direct and indirect impacts of disease response phenotypes on chicken coccidiosis epidemiology: A modelling approach

PONE-D-25-38639R2

Dear Dr. Marie ITHURBIDE,

We’re pleased to inform you that your manuscript has been judged scientifically suitable for publication and will be formally accepted for publication once it meets all outstanding technical requirements.

Kind regards,

Sanaullah Sajid, M.Phil/PhD

Academic Editor

PLOS One
---

## [Editor Report · Acceptance letter]

PONE-D-25-38639R2

PLOS One

Dear Dr. ITHURBIDE,

I'm pleased to inform you that your manuscript has been deemed suitable for publication in PLOS One. Congratulations! Your manuscript is now being handed over to our production team.

Kind regards,

on behalf of

Dr. Sanaullah Sajid

Academic Editor

PLOS One